# Biosynthetic gene cluster profiling predicts the positive association between antagonism and phylogeny in *Bacillus*

Liming Xia [1,3], Youzhi Miao[1,3], A'li Cao [1], Yan Liu[1], Zihao Liu[1], Xinli Sun[1], Yansheng Xue[1], Zhihui Xu[1], Weibing Xun [1], Qirong Shen [1], Nan Zhang [1✉] & Ruifu Zhang [2✉]

Understanding the driving forces and intrinsic mechanisms of microbial competition is a fundamental question in microbial ecology. Despite the well-established negative correlation between exploitation competition and phylogenetic distance, the process of interference competition that is exemplified by antagonism remains controversial. Here, we studied the genus *Bacillus*, a commonly recognized producer of multifarious antibiotics, to explore the role of phylogenetic patterns of biosynthetic gene clusters (BGCs) in mediating the relationship between antagonism and phylogeny. Comparative genomic analysis revealed a positive association between BGC distance and phylogenetic distance. Antagonistic tests demonstrated that the inhibition phenotype positively correlated with both phylogenetic and predicted BGC distance, especially for antagonistic strains possessing abundant BGCs. Mutant-based verification showed that the antagonism was dependent on the BGCs that specifically harbored by the antagonistic strain. These findings highlight that BGC-phylogeny coherence regulates the positive correlation between congeneric antagonism and phylogenetic distance, which deepens our understanding of the driving force and intrinsic mechanism of microbial interactions.

[1] Jiangsu Provincial Key Lab of Solid Organic Waste Utilization, Jiangsu Collaborative Innovation Center of Solid Organic Wastes, Educational Ministry Engineering Center of Resource-saving fertilizers, Nanjing Agricultural University, 210095 Nanjing, Jiangsu, P. R. China. [2] Key Laboratory of Microbial Resources Collection and Preservation, Ministry of Agriculture, Institute of Agricultural Resources and Regional Planning, Chinese Academy of Agricultural Sciences, 100081 Beijing, P. R. China. [3] These authors contributed equally: Liming Xia, Youzhi Miao. ✉email: nanzhang@njau.edu.cn; zhangruifu@caas.cn

Microbes are naturally surrounded by taxonomically different ones with which they compete for scarce resources and space[1]. Competition between different species is generally categorized as exploitation competition, which involves the rapid consumption of a limited resource[2,3], and interference competition, which refers to direct antagonistic interactions[4]. Microbial antagonism is driven by diverse toxins, such as broad-spectrum antibiotics and strain-specific bacteriocins[5,6], and is recognized as a key element regulating populations and determining their success within diverse communities[7]. The stunning diversity of both the categories and functions of antimicrobial metabolites among different species, results in the extraordinary complexity of interference competition between microbes with different phylogenetic relationships[4,8]. Accordingly, illustrating the driving forces and mechanisms of antagonistic competition is crucial for understanding and predicting microbial behaviors during community assemblage and succession[2].

Phylogenetic relatedness is considered to be closely associated with microbial competition, based on its determination of both primary and secondary metabolic profiles[2,4]. Despite the well-established negative correlation between exploitation competition and phylogenetic distance[2], the process of interference competition is much more complicated and remains controversial. Russel et al. demonstrated that inhibition was more prevalent between closely related bacteria, and this negative correlation between antagonism and phylogeny was mediated by the overlap of the metabolic niche among different strains[9]. Conversely, other studies examined congeneric competition in *Vibrio* and *Streptomyces* and revealed that closely related strains competed less than phylogenetically distant strains, which was probably caused by the effect of the prior coexistence and distribution of secondary metabolites in different genomes[7,10,11]. Additionally, the positive relationship between kin discrimination and phylogeny was indicated within *Bacillus subtilis*, which was modulated by genes involved in antimicrobials and cell-surface modifiers[12,13]; however, this correlation was lost or even became a certain extent negative when more distantly *Bacillus* strains were tested for antagonism, probably being dependent on the demand of protecting public goods[14]. Taken together, the relationship between antagonism and phylogenetic distance with regard to microbes from different taxonomical scales or groups, as well as the involved biological mechanism, is still under debate, which limits both our understanding and application of these microbial interactions.

Biosynthetic gene clusters (BGCs) are responsible for the production of various secondary metabolites that contribute to interference competition between different microbes[15,16], and also usually provide resistance against the self-produced antibiotic to protect the host cell[4,17,18]. Although the relevance (or lack of relevance) of BGCs to antagonism or phylogeny has been evaluated in diverse microbes[11,16,17,19–21], the involvement of BGCs profile in mediating the relationship between interference competition and phylogeny, has not been well addressed. Here, we hypothesize that the correlation between BGC and phylogenetic distance can predict the pattern of congeneric antagonism among different taxonomic groups, as strains possessing higher BGC similarity should have a lower probability of inhibiting each other. To test this hypothesis, we referred to the Gram-positive *Bacillus* as the target genus, which is a commonly recognized producer of multifarious secondary metabolites[16,19,22–25], including nonribosomal lipopeptides (e.g., surfactin, iturin, and fengycin families produced by various species)[23], nonribosomal polyketides (e.g., difficidin and macrolactin produced by *B. amyloliquefaciens* and *B. velezensis*)[19], peptide-polyketide hybrid compound (e.g., zwittermicin produced by *B. cereus* and *B. thuringiensis*)[24], and ribosomally synthesized and post-translationally modified

peptides (RiPPs; e.g., lichenicidin produced by *B. licheniformis*)[26]. Based on comparative genomic analysis, antagonistic assessments, and mutant-based verification, we show that the distribution profile of BGCs within *Bacillus* genomes is consistent with their phylogenetic relationship; accordingly, the congeneric antagonism among *Bacillus* strains positively correlates with phylogenetic distance, and this inhibition is dependent on the BGCs that specifically harbored by the antagonistic strain. Our results demonstrate that BGC-phylogeny coherence regulates the positive correlation between congeneric antagonism and phylogenetic distance.

## Results

**Positive correlation between biosynthetic gene cluster (BGC) and phylogenetic distance in the genus *Bacillus*.** BGCs are responsible for the synthesis of secondary metabolites involved in microbial interference competition. To investigate the relationship between BGC and phylogenetic distance within the genus *Bacillus*, we collected 4268 available *Bacillus* genomes covering 139 species from the NCBI database (Supplementary Data 1). Phylogenetic analysis based on the sequences of 120 ubiquitous single-copy proteins[27] showed that the 139 species could be generally clustered into four clades (Fig. 1 and Supplementary Data 2; the phylogenetic tree including all the detailed species information is shown in Supplementary Fig. 1), including a *subtilis* clade that includes species from diverse niches and can be further divided into the *subtilis* and *pumilus* subclades, a *cereus* clade that contains typical pathogenic species (*B. cereus, B. anthracis, B. thuringiensis*, etc.), a *megaterium* clade, and a *circulans* clade.

Prediction using the bioinformatic tool antiSMASH[15] detected 49,671 putative BGCs in the 4268 genomes, corresponding to an average of 11.6 BGCs per genome (Supplementary Data 3). The *subtilis* clade had the most BGCs, 13.1 BGCs per genome (Fig. 2a); the *subtilis* subclade especially accommodates a high abundance of BGCs as 13.6 per genome (Supplementary Fig. 2a), which corresponds to their adaptation in diverse competitive habitats such as plant rhizosphere. The *cereus* and *megaterium* clades possessed moderate number of BGCs as 11.7 and 7.4 per genome, respectively; while the *circulans* clade only had 4.3 BGCs/genome (Fig. 2a and Supplementary Table 1), suggesting a distinct physiological feature and niche adaptation strategy. The two most abundant BGC classes were nonribosomal peptide-synthetase (NRPS) and RiPPs, which had an abundance of 3.7 and 3.1 per genome on average, respectively (Supplementary Fig. 2b and Supplementary Table 1). Interestingly, *subtilis* clade accommodated significantly higher abundance of BGCs in another polyketide synthase (PKSother; 2.0 per genome vs. 0.0–1.1 per genome) and PKS-NRPS Hybrids (0.7 vs. 0.0–0.2) classes, as compared with the three other clades (Supplementary Table 1); while *cereus* clade had more BGCs in RiPPs than other clades on average (Supplementary Table 1). Overall, the profile of BGC products and classification was generally consistent with the phylogenetic tree (Supplementary Fig. 3).

To further evaluate whether the diversity and concrete distribution of the BGCs among genomes were relevant to the phylogenetic relatedness, we selected 545 representative *Bacillus* genomes based on the following criteria: (i) high genome sequencing quality that available for further BGC distance calculation; (ii) covering all *Bacillus* species; (iii) for each species, the representative genome(s) was/were chosen from each of the main branches in the phylogram of genomes within this species. We analyzed the interactive sequence similarity network of BGCs in these genomes by using the "biosynthetic gene similarity clustering and prospecting engine" (BiG-SCAPE)[28]. The (dis)

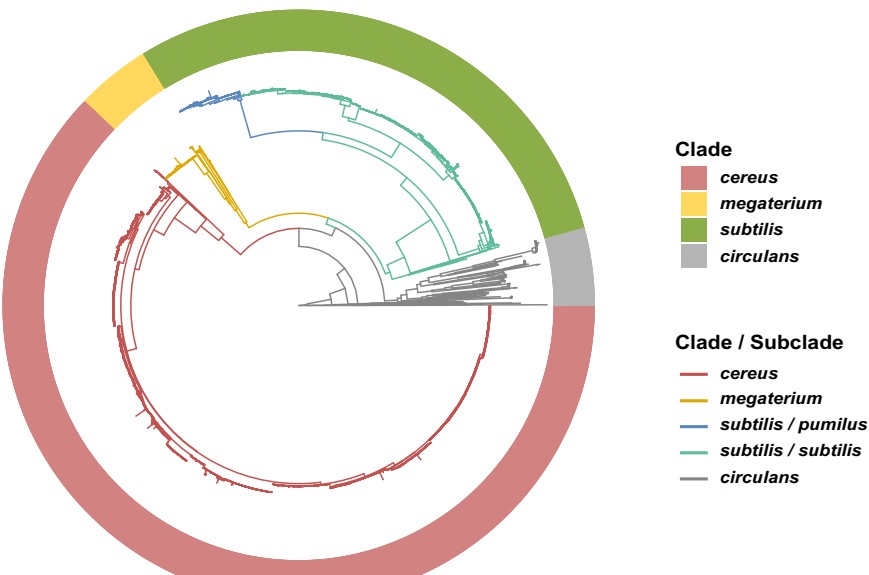

**Fig. 1 Phylogram of the tested *Bacillus* genomes.** The maximum likelihood (ML) phylogram of 4268 *Bacillus* genomes was based on the sequences of 120 ubiquitous single-copy proteins[27]. The phylogenetic tree shows that *Bacillus* species can be generally clustered into the *subtilis* (light green circle; further includes *subtilis* (dark green) and *pumilus* (blue) subclades as shown in the branches), *cereus* (red), *megaterium* (yellow), and *circulans* (gray) clades. For detailed information of the species, please refer to the phylogenetic tree in Supplementary Fig. 1.

similarity of paired BGCs was calculated based on a combination of three metrics, including the Jaccard index (JI), adjacency index (AI), and domain sequence similarity (DSS), which resulted in 1110 gene cluster families (GCFs) and 76 gene cluster clans (GCCs) of the 4877 putative BGCs (Supplementary Data 4 and 5). The hierarchal clustering based on the abundance of these GCFs among each genome (Supplementary Data 6) indicated that each phylogenetic clade/subclade revealed its own distinctive BGCs distribution profile, and possessed a number of taxonomy-specific secondary metabolites (Fig. 2b, Supplementary Fig. 4, and Supplementary Data 4). The widespread BGCs in *cereus* clade included fengycin, bacillibactin, bacteriocin, NRPS, and petrobactin, in which petrobactin was a clade-specific BGC; polyoxypeptin, thurincin, and zwittermicin were also specific molecules but were mainly present in a certain of *B. cereus* and *B. thuringiensis* genomes (Fig. 2b and Supplementary Fig. 4). In *subtilis* clade, most species possessed surfactin, fengycin, bacilysin, bacillibactin, and T3PKS, while each group can also produce unique BGCs, such as betalactone for *B. pumilus*, subtilin and subtilosin for *B. subtilis*, difficidin and macrolactin for *B. amyloliquefaciens* and *B. velezensis*, mersacidin, plantazolicin, and plipastatin for a certain of genomes in the above two species, and lichenysin for *B. licheniformis* (Fig. 2b and Supplementary Fig. 4). Species in *megaterium* clade mostly accommodated siderophore, surfactin, and T3PKS, some strains can potentially produce lanthipeptide, paeninodin, or bacteriocins. The dominating BGC in *circulans* clade was identified as T3PKS, and some species may synthesize siderophore, bacteriocin, or lanthipeptide (Fig. 2b and Supplementary Fig. 4).

Furthermore, we calculated the BGC distance between different genomes based on the above GCFs clustering data, and found a significant positive correlation between the BGC and phylogenetic distance (Fig. 2c) ($P < 2.2 \times 10^{-16}$, $R^2 = 0.2847$). Genomes of phylogenetically close relatives (phylogenetic distance < 0.1) carried highly similar arsenal of BGCs (Fig. 2c and Supplementary Fig. 5). In more distantly related species (phylogenetic distance > 0.3) the BGCs distance increased with phylogenetic distance but some BGCs were also highly dispersed (Fig. 2c and Supplementary Fig. 5b). This suggests that some BGC loci are

shared between different clades (Supplementary Fig. 5b), for example, between *subtilis-cereus* clades or *circulans* and other clades. To summarize, these findings demonstrate that the BGCs distribution profile was generally dependent on the phylogenetic relationship within the genus *Bacillus*.

**Antagonism positively correlates with both the phylogenetic and BGC distance in *Bacillus*.** BGCs not only contribute to the synthesis of secondary metabolites but also usually afford self-tolerance to the antibiotic they encode[29]. We therefore hypothesized that the BGC-phylogeny coherence in *Bacillus* (Fig. 2b, c) determines a positive correlation between antagonism and phylogenetic distance. To verify this hypothesis, we first used the bacterial colony confrontation assay to investigate the relationship between the antagonistic efficiency and phylogenetic distance of the paired strains (Supplementary Fig. 6). The antagonistic bacteria includes eight heterogenic strains from the *subtilis* or *cereus* clade (two from *subtilis* subclade, two from *pumilus* subclade, and four from *cereus* clade), which are the two dominant groups within the genus *Bacillus* and have been explored to provide abundant secondary metabolites (Fig. 2a); the target bacteria includes 59 strains covering all four clades, and their species were located on the main branches of the *Bacillus* phylogenetic tree (Supplementary Data 7 and Supplementary Fig. 1). The results indicated that all of these antagonistic strains tended to show stronger antagonistic ability towards distant species than towards closely related species. For instance, *B. amyloliquefaciens* ACCC19745 and *B. pumilus* ACCC04450 (both belong to the *subtilis* clade) showed weak antagonism against strains in the same subclade but exhibited an increased antagonistic ability to the out-clade species (Fig. 3a, b, $P = 9.30 \times 10^{-5}$ and $P = 9.68 \times 10^{-5}$, respectively). Correspondingly, the antagonistic abilities of *B. thuringiensis* YX7 and *B. mobilis* XL40 (*cereus* clade) towards other *Bacillus* strains were also enhanced with increasing phylogenetic distance (Fig. 3c, d, $P = 0.007$ and $P = 0.008$, respectively). Based on the results of the colony confrontation assays, a significant positive correlation between antagonism and phylogenetic distance was revealed (Fig. 3e,

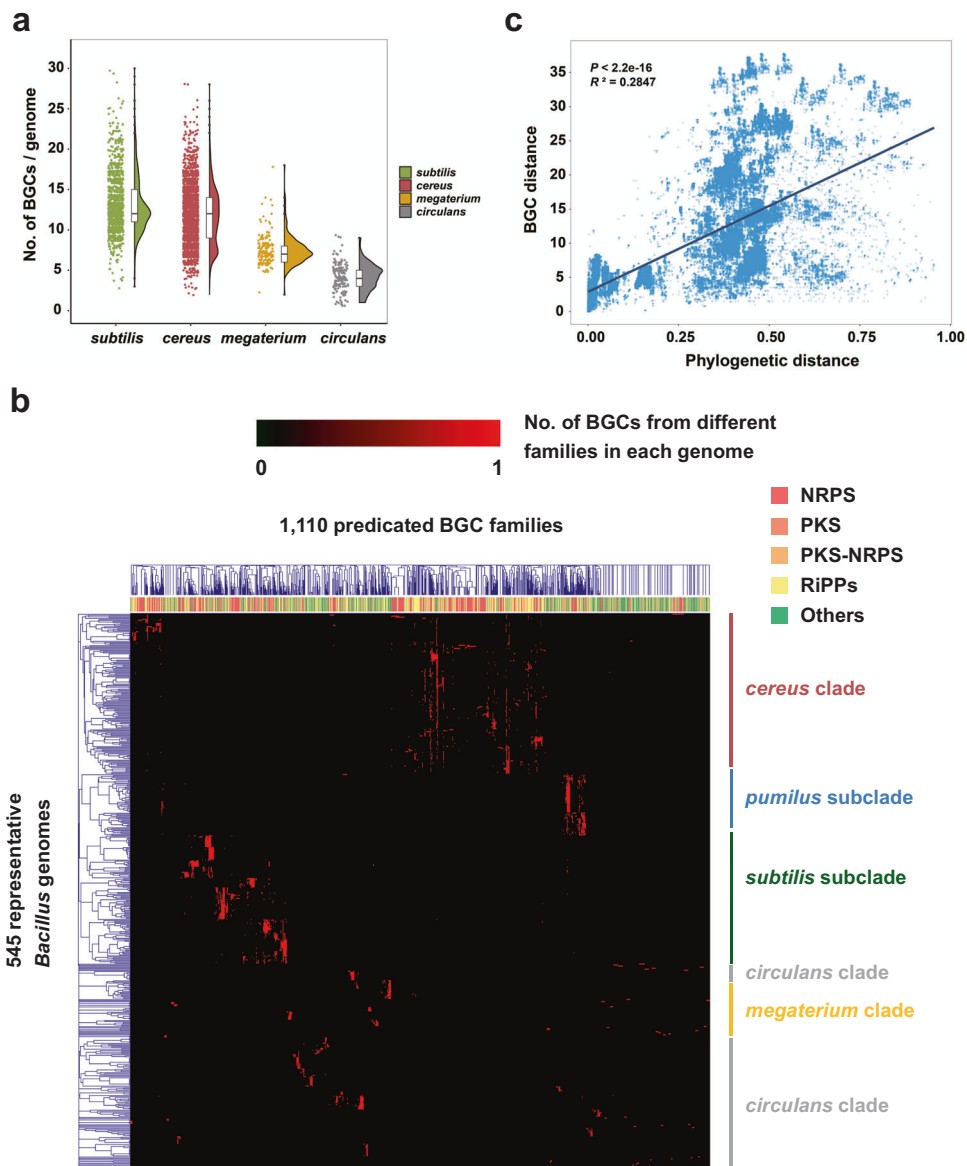

**Fig. 2 Biosynthetic gene cluster (BGC) distribution is correlated with phylogeny in the genus *Bacillus*. a** The numbers of BGCs in the 4268 *Bacillus* genomes from different clades as defined by antiSMASH[15]. In the violin plot, the centre line represents the median, violin edges show the 25th and 75th percentiles, and whiskers extend to 1.5× the interquartile range. **b** Hierarchal clustering among the 545 representative *Bacillus* genomes based on the abundance of the different biosynthesis gene cluster families (GCFs). Each column represents a GCF, which was classified through BiG-SCAPE by calculating the Jaccard index (JI), adjacency index (AI), and domain sequence similarity (DSS) of each BGC[28]; the color bar on the top of the heatmap represents the BGC class of each GCF, where PKS includes classes of PKSother and PKSI, PKS-NRPS means PKS-NRPS Hybrids, Others includes classes of saccharides, terpene, and others. Each row represents a *Bacillus* genome, and the abundance of each GCF in different genomes is shown in the heatmap. The left tree was constructed based on the distribution pattern of GCFs, which showes a similar pattern to the phylogram in Fig. 1. **c** The correlation between the BGC and phylogenetic distance of the 545 representative *Bacillus* genomes ($P < 2.2 \times 10^{-16}$, $R^2 = 0.2847$). Linear model (LM) was used for the correlation analysis and adjustments were made for $R^2$ calculation; one-sided F test was applied for multiple comparisons (F-statistic: $5.669 \times 10^4$ on 1 and 142,436 DF). Source data are provided as a Source Data file.

$P = 7.338 \times 10^{-15}$, $R^2 = 0.1427$). To further clarify whether this association was mediated by the BGC distribution pattern, we calculated the predicted BGC distance among all the tested paired strains (for strains whose genomes have not been completely sequenced, we referred to the *Bacillus* genomes in the NCBI database that shared the highest 16 S rRNA similarity). Interestingly, there was also a significant positive correlation between antagonism and the predicted BGC distance (Fig. 3f, $P = 1.971 \times 10^{-11}$, $R^2 = 0.1076$), suggesting that BGC profiling is likely to play a role in regulating interspecies antagonism.

Furthermore, to check the positive antagonism-phylogeny correlation in a more defined system, we performed a fermentation supernatant assessment to test the inhibition between paired strains. This strategy can avoid potential bacterial nutrient competition and is feasible for a wider range of antagonistic strains since antagonism is not influenced by the growth speed or morphology of the colony (Supplementary Fig. 6). Here, we expanded the antagonistic strains to 17 species covering all four phylogenetic clades (Supplementary Data 7). The extracellular metabolites of antagonistic bacteria also generally showed

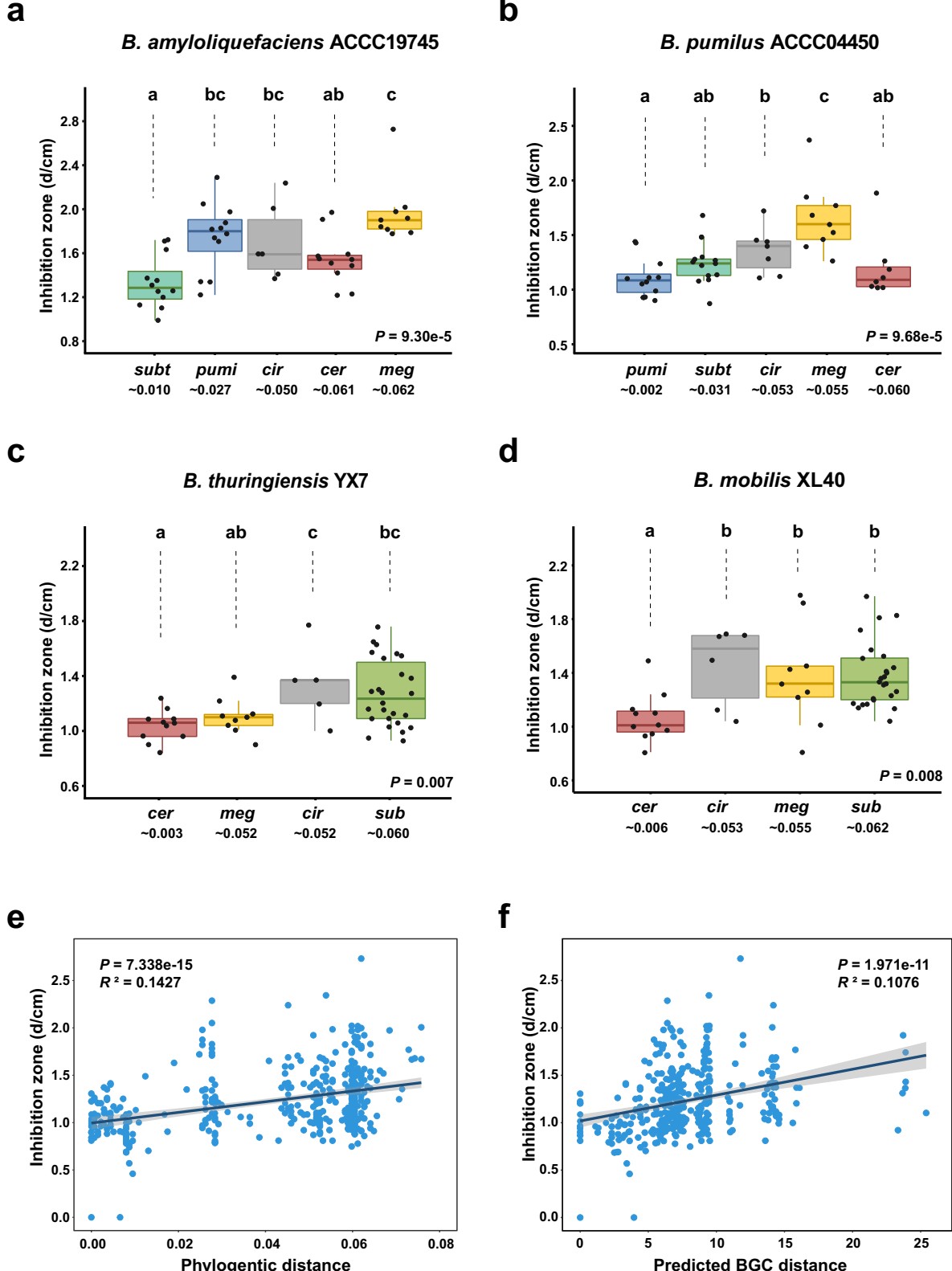

stronger inhibition to distantly related strains than to closely related strains; this pattern was particularly clear for antagonistic strains in the *subtilis* clade, which harbored the most BGCs (Fig. 4a and Supplementary Fig. 6). Interestingly, *B. ginsengihumi* ACCC05679 was found to be highly antagonistic against nearly all tested strains (Fig. 4a), which might be attributed to the synthesis of β-lactone based on prediction using its draft genome

(Supplementary Data 7), a natural product that can inhibit diverse Gram-positive bacteria[30]. As expected, antagonism showed a significant positive correlation with both phylogenetic (Fig. 4b, $P < 2.2 \times 10^{-16}$, $R^2 = 0.1618$) and predicted BGC distance (Fig. 4c, $P = 2.241 \times 10^{-13}$, $R^2 = 0.08523$); while within clades, this significant concordance between inhibition and phylogram was recovered in the *subtilis* clade but not in

**Fig. 3 Colony antagonism phenotype is positively correlated with the phylogenetic and BGC distance within *Bacillus* species. a–d** Inhibition of colonies of *B. amyloliquefaciens* ACCC19745, *B. pumilus* ACCC04450, *B. thuringiensis* YX7, and *B. mobilis* XL40 against *Bacillus* from different clades. The number below the abbreviations indicates the average 16 S rRNA gene phylogenetic distance of the target strains with the corresponding antagonistic strain. Each inhibition assay includes three biological replicates and the average is shown as points in the boxplots ($n = 50, 49, 51$, and 51 for **a**, **b**, **c**, and **d**, respectively) and used for the correlation analysis. In all boxplots, the centre line represents the median, box edges show the 25th and 75th percentiles, and whiskers extend to 1.5× the interquartile range; boxplots with different letters are statistically different according to the two-sided Duncan's multiple range tests ($P < 0.05$), where the significance for **a**, **b**, **c**, and **d** are $9.30 \times 10^{-5}$, $9.68 \times 10^{-5}$, 0.007, and 0.008, respectively. **e**, **f** Correlation between the antagonism phenotype (diameter of the inhibition zone) and 16 S rRNA phylogenetic distance (**e**, F-statistic: 65.58 on 1 and 387 DF, $P = 7.338 \times 10^{-15}$, $R^2 = 0.1427$) or predicted BGC distance (**f**, F-statistic: 47.79 on 1 and 387 DF, $P = 1.971 \times 10^{-11}$, $R^2 = 0.1076$) among all the tested paired strains. For strains whose genomes have not been completely sequenced, we referred to the *Bacillus* genomes in the NCBI database that shared the highest 16 S rRNA similarity. The error bands indicate the 95% confidence intervals. Linear model (LM) was used for the correlation analysis and adjustments were made for $R^2$ calculation; one-sided F test was applied for multiple comparisons. Source data are provided as a Source Data file.

other three clades (Supplementary Fig. 7; *subtilis* clade, $P = 6.337 \times 10^{-12}$, $R^2 = 0.3469$; *cereus* clade, $P = 0.5752$, $R^2 = -0.01873$; *megaterium* clade, $P = 0.985$, $R^2 = -0.06664$; *circulans* clade, $P = 0.09022$, $R^2 = 0.1335$). Intriguingly, we found the correlation between antagonism and phylogenetic distance for each individual antagonistic strain, was positively associated with the predicted quantity of BGCs in this bacteria (for strains whose genomes have not been completely sequenced, we referred to the average quantity of BGCs in this species; Fig. 4d, $P = 0.005992$, $R^2 = 0.3657$). This finding suggests that the antagonistic bacteria carrying abundant BGCs (e.g., >10) tend to have a clear positive correlation between inhibition phenotype and phylogenetic distance, while those with fewer BGCs (e.g., <8) show a weak or even irregular antagonistic pattern against diverse targets. Furthermore, the BGC-phylogeny coherence was similar among all the antagonistic strains (no significant relevance with BGCs No.; Supplementary Fig. 8, $P = 0.4201$, $R^2 = -0.01994$), while the antagonism-BGC distance correlation revealed a positive association with the quantity of BGCs in bacteria (i.e., bacteria having fewer BGCs showed a weak antagonism-BGC distance relevance, and vice versa.; Fig. 4e, $P = 0.0494$, $R^2 = 0.1824$), which can partially explain the weak correlation between antagonism and phylogenetic distance in strains possessing fewer BGCs.

**The positive correlation of antagonism and phylogenetic distance in *Bacillus* is mediated by the specifically harbored BGCs in antagonistic strains.** Having determined that the positive correlation between antagonism and phylogenetic distance was consistent with the BGC-phylogeny coherence in *Bacillus*, we further investigated the mediation mechanism of BGCs in the interspecies interactions. We used a typical secondary metabolite producer, *B. velezensis* SQR9 belonging to *subtilis* clade, to identify the primary antagonistic antibiotic towards different strains. Strain SQR9 devotes ~9.9% of its genome to the synthesis of various antimicrobial metabolites[31], including five non-ribosomal lipopeptides or dipeptides (surfactin, bacillomycin D, fengycin, bacillibactin, and bacilysin), three polyketides (macrolactin, bacillaene, and difficidin)[32], and one antimicrobial fatty acid (FA; bacillunoic acid)[18]. The antagonistic characteristics of SQR9 mutants deficient in each of the above BGCs and SQR9Δ*sfp* with the 4'-phosphopantetheinyl transferase gene deleted and only bacilysin can be synthesized[22,31] (Supplementary Data 7), towards 24 target strains (Supplementary Data 7) were investigated using a fermentation supernatant inhibition assay. Generally, the inhibition phenotype was almost completely attributed to these given BGCs (Fig. 5, the "sum" column). SQR9Δ*sfp* nearly completely lost its antagonism towards all the target strains, suggesting that the synthesis of the antibiotics involved in congeneric antagonism is strongly dependent on Sfp (Fig. 5). The active antimicrobial metabolites were found to be relevant to the phylogenetic positions of the target strains, as a specific BGC was

involved in the inhibition of strains in one taxonomic group. In detail, difficidin dominated the suppression of the *megaterium* clade (Fig. 5); macrolactin was the primary antibiotic against the *cereus* clade (Fig. 5); difficidin and bacillaene both contributed to the inhibition of the *circulans* clade (Fig. 5). Strain SQR9 also used the antimicrobial FA to compete with strains in closely related species (*B. halotolerans* CF7, *B. licheniformis* LY2, and *B. sonorensis* YX13) (Fig. 5). Furthermore, we assigned the BGCs presence in the testing strains based on their genomic information, or for the non-sequenced strains if more than 80% of the corresponding *Bacillus* species genomes possessed the cluster (marked by the cross in Fig. 5). Importantly, antagonism was fully attributed to the BGCs that were present in strain SQR9 but absent in the target strains, while the metabolites shared by both SQR9 and the target strain were not involved. Additionally, the three identified functional antibiotics (difficidin, macrolactin, and bacillaene) for congeneric inhibition belongs to PKSother or PKS-NRPS Hybrids classes; enrichment of both classes in *subtilis* clade than the other three (Supplementary Table 1) coincides with the weak inhibition on *subtilis* clade by strain SQR9. Taken together, these results demonstrated that interference competition was dependent on the BGCs specifically harbored by the antagonistic strains, and strains that shared more (analogous) BGCs tended to have a lower probability and intensity of antagonism. It should be noted that not all the unique BGCs in antagonistic strains contributed to the inhibition of the target (Fig. 5) and the outcome may also depend on whether a specific secondary metabolite is being synthesized or not, this aspect still needs further investigation.

## Discussion
The competition-relatedness hypothesis proposed by Charles R. Darwin in the *Origin of Species*, that is, congeneric species are likely to compete more fiercely by means of their functional similarity[33], has been examined in various organisms and has received both positive and negative support[34–37]. With regard to the microbial fierce competition exemplified by antagonism, Russel et al. found a negative correlation between inhibition probability and phylogenetic distance[9]; some other scientists discovered a positive relationship between antagonistic interaction (including kin discrimination) and phylogenetic dissimilarity in genus *Vibrio*[7] and *Streptomyces*[10,11], and species *B. subtilis*[12,13]. The present study demonstrated that antagonism tended to be positively correlated with phylogenetic distance within the genus *Bacillus* (Figs. 3 and 4). Comprehensively, we pronounce that within a relatively narrow phylogenetic range such as for congeneric strains, BGC similarity, which determines the secondary metabolite profile, appears to be the main driver of antagonistic interactions (Figs. 2–5). BGCs themselves, or other elements in the same genome, usually encode antibiotic resistance genes (ARGs) to afford self-resistance by providing active efflux or

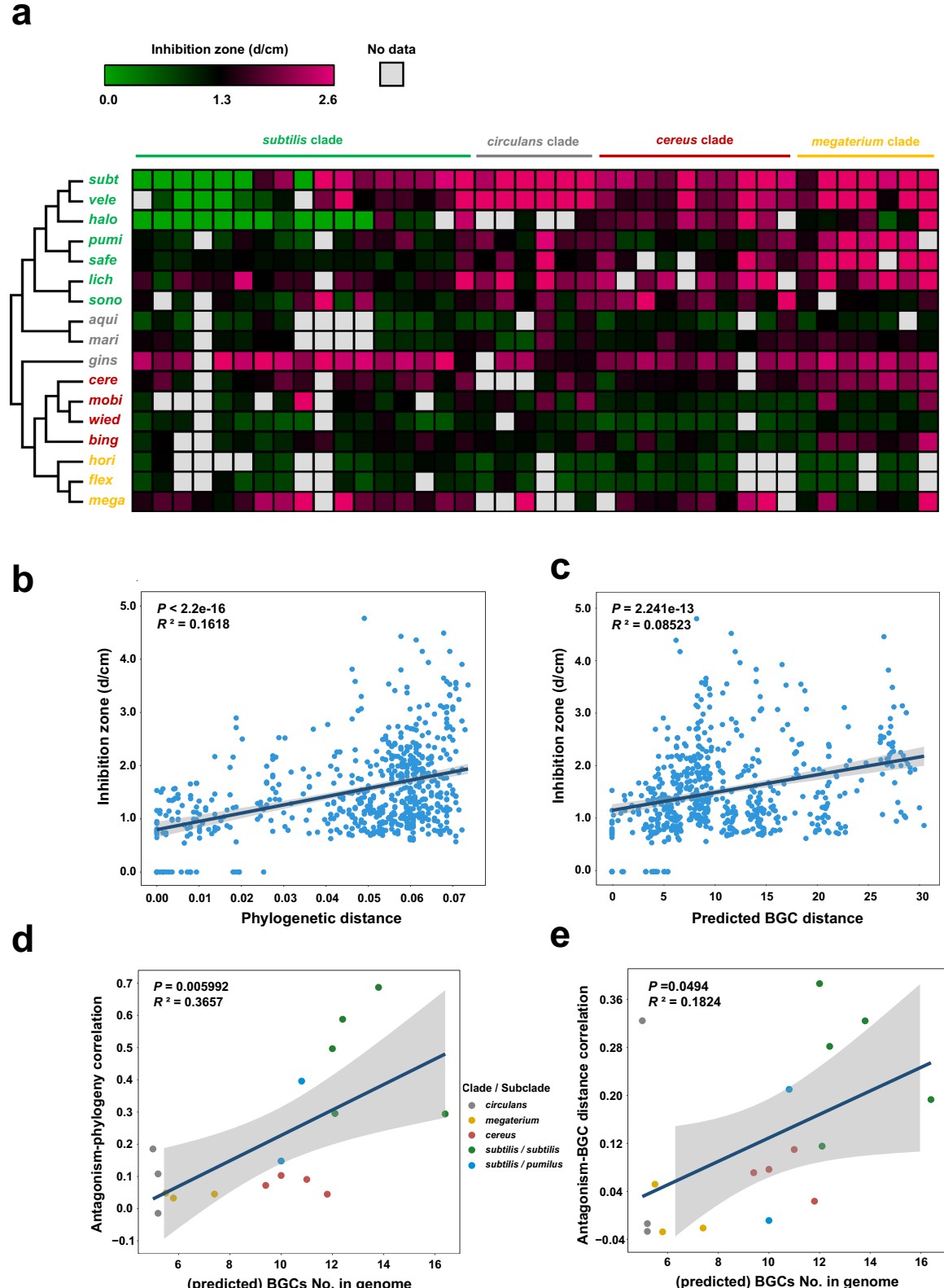

modification of the relevant metabolite[29,38], while the absence of a specific BGC suggests the potential to be sensitive to this compound[17,18]; this principle was also confirmed by the observation that antagonism was dependent on the BGC that was present in the antagonistic strain but absent in the target strain (Fig. 5). In addition, despite the mobility of BGCs among different microbes[16,39] as indicated by the prevalence of a number of

homologous clusters in genomes from phylogenetically distant species (Supplementary Fig. 9), their distribution pattern was generally in accordance with the phylogenetic relationship within genus (Fig. 2b, c). Consequently, closely related species with a higher BGC similarity have a lower probability of inhibiting each other, while distant species in the same genus are likely to suppress each other more fiercely (Figs. 3 and 4). Therefore, we

**Fig. 4 Congeneric inhibition by fermentation supernatants is positively correlated with the phylogenetic and BGC distance in *Bacillus*. a** Heatmap showing the antagonistic profiles of the fermentation supernatant of 17 antagonistic strains (in the left column) on the 40 target strains (in the top line). The maximum likelihood (ML) phylogenetic tree was constructed based on the 16 S rRNA sequence of the 17 antagonistic strains: subt, *B. subtilis* RZ30; vele, *B. velezensis* SQR9; halo, *B. halotolerans* CF7; pumi, *B. pumilus* ACCC04450; safe, *B. safensis* LY9; lich, *B. licheniformis* CC11; sono, *B. sonorensis* YX13; aqui, *B. aquimaris* XL39; mari, *B. marisflavi* XL37; gins, *B. ginsengihumi* ACCC05679; cere, *B. cereus* ACCC10263; mobi, *B. mobilis* XL40; wied, *B. wiedmannii* XL36; bing, *B. bingmayongensis* KF27; hori, *B. horikoshii* ACCC02299; flex, *B. flexus* DY11; mega, *B. megaterium* ACCC01509. Each inhibition assay includes three biological replicates and the average is shown in the heatmap and used for the correlation analysis. **b, c** Correlation between the antagonism phenotype (diameter of the inhibition zone) and 16 S rDNA phylogenetic distance (**b**, F-statistic: 115.6 on 1 and 593 DF, $P < 2.2 \times 10^{-16}$, $R^2 = 0.1618$) or the predicted BGC distance (c, F-statistic: 56.34 on 1 and 593 DF, $P = 2.241 \times 10^{-13}$, $R^2 = 0.08523$) among all the tested paired strains. For strains whose genomes have not been completely sequenced, we referred to the *Bacillus* genomes in the NCBI database that shared the highest 16 S rRNA similarity. **d, e** Correlation between the antagonism-phylogeny association (**d**, F-statistic: 10.23 on 1 and 15 DF, $P = 0.005992$, $R^2 = 0.3657$) or antagonism-BGC distance association (**e**, F-statistic: 4.57 on 1 and 15 DF, $P = 0.0494$, $R^2 = 0.1824$) and the (predicted) quantity of BGCs in antagonistic strains. For strains whose genomes have not been completely sequenced, we referred to the average quantity of BGCs in this species. The color of the dots represents the clade/subclade which the antagonistic strains belong to. The error bands indicate the 95% confidence intervals. Linear model (LM) was used for the correlation analysis and adjustments were made for $R^2$ calculation; one-sided F test was applied for multiple comparisons. Source data are provided as a Source Data file.

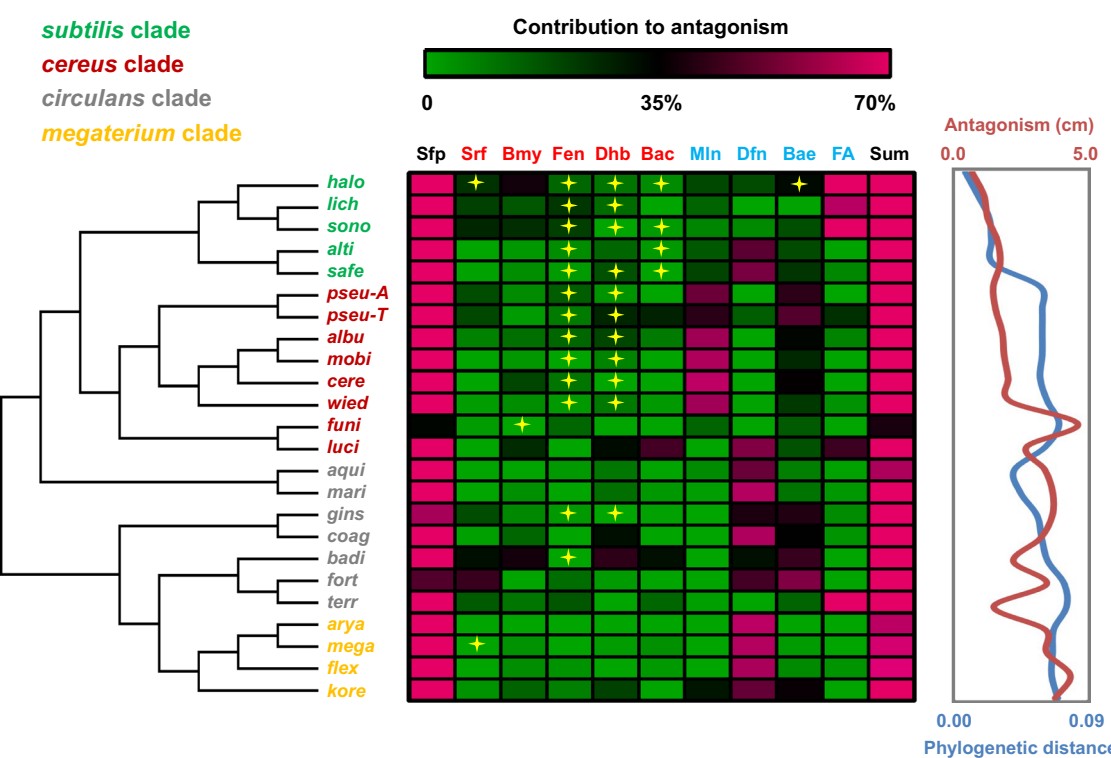

**Fig. 5 Contribution of BGCs to antagonizing *Bacillus* species from different clades by *B. velezensis* SQR9.** The heatmap shows the contribution of each BGC product (on the top) to the inhibition of each target strain (on the left), which was calculated as the percentage of the decreased inhibition zone of the corresponding BGC-deficient mutant compared with wild-type. The maximum likelihood (ML) tree on the left was constructed based on the 16 S rRNA sequences of the 24 target strains: halo *B. halotolerans* CF7, lich *B. licheniformis* LY2, sono *B. sonorensis* YX13, alti *B. altitudinis* LY37, safe *B. safensis* LY9, pseu-A *B. pseudomycoides* ACCC10238, pseu-B *B. pseudomycoides* TZ8, albu *B. albus* XL388, mobi *B. mobilis* XL40, cere *B. cereus* ACCC10263, wied *B. wiedmannii* CF23, funi *B. funiculus* ACCC05674, luci *B. luciferensis* XL165, aqui *B. aquimaris* XL39, mari *B. marisflavi* XL37, gins *B. ginsengihumi* ACCC05679, coag *B. coagulans* ACCC10229, badi *B. badius* ACCC60106, fort *B. fortis* ACCC10219, terr *B. terrae* TL19, arya *B. aryabhattai* XL26, mega *B. megaterium* ACCC01509, flex *B. flexus* DY11, kore *B. koreensis* ACCC05681. The abbreviations on the top (except Sfp) represent different BGC products: Srf surfactin, Bmy bacillomycin D, Fen fengycin, Dhb bacillibactin, Bac bacilysin, Mln macrolactin, Bae bacillaene, Dfn difficidin, FA an antimicrobial fatty acid, bacillunoic acid. Specifically, Sfp (phospho-pantheinyltransferase) is not an antibiotic but is necessary for modification of the above antibiotics and ensuring their activity, except for bacilysin[22,30]; here, the contribution of Sfp to antagonism means the relative reduction of inhibition by SQR9Δ*sfp* compared to that of wild-type SQR9 towards different targets. "Sum" represents the overall contribution of the nine BGCs (excludes Sfp) to the antagonism against different targets. The cross represents the BGCs presence in the testing strains based on their genomic information, or for the non-sequenced strains if more than 80% of the corresponding *Bacillus* species genomes possessed the cluster. The curves in the right box show the antagonistic phenotype and phylogenetic distance of *B. velezensis* SQR9 with each of the target strains. Each inhibition assay included six biological replicates and the average contribution was shown in the heatmap. Source data are provided as a Source Data file.

highlighted that the coherence between BGCs distribution and phylogenetic characteristics is one of the crucial factors regulating congeneric interactions. Comparatively, at a larger taxonomic scale, the significant variation in the BGCs distribution among bacteria from different genera or even phyla can contribute to an irregular correlation between the secondary metabolite profile and phylogenetic distance[28,40,41]. In this situation, functional similarity, such as metabolic niche overlap, may become the main driver and lead to a negative correlation between antagonism and phylogenetic distance[9]. Taken together, this taxonomic range-dependent association between antagonism and phylogeny should indicate a cooperation-competition tradeoff during microbial interactions and is coordinated by a set of sophisticated molecular mechanisms.

Interestingly, the positive correlation between antagonism and phylogeny was relatively strong in antagonistic strains possessing abundant BGCs (e.g., >10), but was weak or even not significant in those harboring fewer BGCs (e.g., <8) (Fig. 4d). This divergence is likely to arise from the altered antagonism-BGC distance correlation in different bacteria (Fig. 4e and Supplementary Fig. 8). As inhibition is mainly dependent on specific BGCs that are present in antagonistic strain and absent in target strain (Fig. 5), we further identified the unique BGCs in each antagonistic strain for confronting different targets (Supplementary Data 8). It has been shown that for bacteria possessing fewer BGCs (e.g., *B. aquimaris* XL39 and *B. horikoshii* ACCC02299), the low number of the unique antibiotics (usually ≤3, excluding those without antimicrobial activity, e.g., oligosaccharide and phosphonate), was not appropriate for statistical analysis and can impair the biological regularity of inhibition phenotype against diverse targets (Supplementary Data 8). This attribution can partially explain the weak correlation between antagonism and phylogeny in these strains. Furthermore, other potential factors may also contribute to the relatively low correlation between antagonism and phylogeny, for example, the unknown secondary metabolites of bacteriocins may specifically kill close relatives, internal genetic variation (e.g., point mutations, partial deletion, altered gene regulation, and silent expression) or external cues (e.g., environmental factors and competing strains) can affect the antibiotic production[11,17,42,43], and the undiscovered genetic and physiological features may also regulate the response to different predicted BGCs. It would be important to identify more secondary metabolites responsible for bacterial interference competition and to further investigate the exquisite regulation characteristics of these functional molecules.

Noticeably, there are some differences between our finding and that reported by Lyons & Kolter, who demonstrated a negative correlation between kin discrimination and phylogeny[14]. This discrepancy may be attributed to the following reasons: (i) The antagonism in this research were evaluated by colony confrontation and fermentation supernatant inhibition, which was dominated by diffusible secondary antibiotics within a comparatively longer distance; while the kin discrimination in Lyons & Kolter's study was assessed based on swarm interaction, biofilm meeting, and halo formation[14], which was likely to be dependent on closer cell association (e.g., toxin-antitoxin system and cell-surface contact)[13]. (ii) The antagonism-phylogeny correlation in our study was calculated based on interactions between diverse antagonistic and target strains. Comparatively, the halo assay performed previously examined the inhibition of one indicated species (*B. subtilis* NCIB3610) by different testing strains[14]; as discussed above, the different BGCs distribution patterns (e.g., the quantity) among distinct antagonistic strains can influence the inhibition phenotype and its relationship with phylogenetic distance. In general, we consider that Lyons & Kolter's study has provided important knowledge with regard to kin discrimination

and close contact, especially in a mixed bacterial population[14]; and our study focused more on the antagonistic interaction *sensu lato*, which can occur within a long distance.

*Bacilli* possess an amazing capacity to synthesize a diverse range of secondary metabolites; previous studies have indicated the phylogenetic conservation of BGCs in the genus *Bacillus* and identified multiple species/clade-specific molecules[16,17,44]. Based on bioinformatics analysis of genomes from a larger scale, we also demonstrated the phylogenetic dependence of BGCs distribution pattern in the genus *Bacillus* (Fig. 2b, c). Specifically, the *subtilis* clade appeared to be the most abundant arsenal for secondary metabolites[16], where the 1259 genomes possessed 16,502 BGCs belonging to 117 products, 310 GCFs, and 47 GCCs, including numerous distinctive and powerful products such as difficidin and macrolactin that produced by *B. amyloliquefaciens* and *B. velezensis*, lichenicidin produced by *B. licheniformis*, and so on (Supplementary Data 4). On the other hand, the BGC-phylogeny regression suggested that the variation of secondary metabolites among intra- or closely related species was slight and stable, but can be either moderate or drastic among distant groups (Fig. 2c and Supplementary Fig. 5). This finding coincides with the transferability of BGCs among different genomes and acquisition through horizontal gene transfer (HGT)[16,39] (Supplementary Fig. 9); therefore, the correlation ($R^2 = 0.2847$, Fig. 2c) is relatively lower as compared with other conserved genes (e.g., housekeeping genes or metabolism relevant characteristics)[45].

The cooperation-competition tradeoff among microbial interactions is usually the consequence of the balance between benefits, such as public goods sharing and cross-feeding, and costs, such as resource competition and stress resistance[2,14]. Within a narrow phylogenetic range (e.g., congeneric interactions), only closely associated species can share secreted cooperative goods, such as surfactin for swarming and biofilm formation[14] or siderophores for iron acquisition[46,47], while metabolic similarity decreases moderately or even changes irregularly with phylogenetic distance[9,11]. Consequently, closely related species can enjoy a higher benefit of public goods than the cost of resource competition and therefore exhibit a relatively weak antagonistic tendency, while distantly related species can hardly share public goods but still confront nutrient competition; thus, the relatively lower benefit than cost contributes to strong demand for congeneric inhibition. Over a broad phylogenetic range (e.g., intraphylum interactions), all strains can barely exploit mutually cooperative goods; hence, their benefit-cost balance is predominantly driven by the cost of nutrient competition[9]. As a result, metabolic similarity facilitates the negative correlation between antagonism and phylogenetic distance. This dualistic relationship between interference competition and phylogenetic distance could be a consequence of natural selection, which impels microbes to balance cooperation and competition in an economical manner and plays an important role in regulating community assemblage and succession[48].

In conclusion, the present study demonstrates the consistency between the BGCs distribution and phylogenetic tendency within the genus *Bacillus*, and this coherence acts as the main factor driving the positive correlation between congeneric antagonism and phylogenetic distance, especially in strains possessing abundant BGCs. We expect this positive association between congeneric antagonism and phylogeny is either pronounced[10,11] or can be predicted in other genera with abundant BGCs. This study deepens our understanding of the driving forces and intrinsic mechanism of microbial interactions and provides implications for designing synthetic microbial communities and manipulating population assemblages for practical purpose.

## Methods

**Bacterial strains and growth conditions**. All 89 *Bacillus* strains used in this study are listed in Supplementary Data 7, including 3 strains obtained from the *Bacillus* Genetic Stock Center (BGSC), 17 strains obtained from the Agricultural Culture Collection of China (ACCC) that originated from different environmental samples, 59 strains isolated from the plant rhizosphere by this laboratory, and 10 mutants of *B. velezensis* SQR9. All strains were grown at 30 °C in low-salt Luria-Bertani (LLB) medium (10 g L$^{-1}$ peptone; 5 g L$^{-1}$ yeast extract; 3 g L$^{-1}$ NaCl); when necessary, final concentrations of antibiotics were added as follows: 100 mg L$^{-1}$ spectinomycin (Spc) and 20 mg L$^{-1}$ zeocin (Zeo). To collect the fermentation supernatant for antagonism assessment, the bacterial strains were cultured in Landy medium[49]. The 16 S rRNA genes were amplified with the 27 F (5′-AGAGTTTGATCCTGGC TCAG-3′) and 1492 R (5′-GGTTACCTTGTTACGACTT-3′) primers and were subsequently Sanger sequenced. The taxonomic affiliations of these strains were determined through the EzBioCloud database and NCBI BLAST.

**Bacillus genomic, phylogenetic, and biosynthetic gene cluster analysis**. In total, 4268 available genomes from 139 different *Bacillus* species were downloaded from the NCBI database using the ncbi-genome-download script (https://github.com/kblin/ncbigenome-download/) (Supplementary Data 1). Then, a phylogenetic tree was constructed based on the concatenation of 120 ubiquitous single-copy proteins using GTDB-Tk 1.4.1 software with the default parameters[27,50]. The resulting tree was subsequently visualized and edited with Figtree 1.4.4 (http://tree.bio.ed.ac.uk/software/figtree/). Maximum likelihood (ML) phylogenetic trees of the 16 S rRNA sequence of the strains used in antagonism assessments were constructed by MEGA 5.0.

Biosynthetic gene clusters (BGCs) of all 4268 *Bacillus* genomes were predicted using antiSMASH 5.0 software[15]. Considering that numerous congeneric strains shared highly similar BGC profiling, in order to compare the BGC distribution among different *Bacillus* groups more adequately, 545 representative *Bacillus* genomes covering all species with high sequencing quality, were further selected for BGC distance analysis (Supplementary Data 1). In detail, the BGC distances were calculated as the weighted combination of the Jaccard Index (JI), adjacency index (AI), and domain sequence similarity (DSS), which resulted in the classification of different gene cluster families (GCFs) and gene cluster clans (GCCs) based on the two cutoff values (0.3 and 0.7, respectively) by the "biosynthetic gene similarity clustering and prospecting engine" (BiG-SCAPE) software[28]. Each *Bacillus* genome was thus annotated to have different GCFs or GCCs, forming a matrix table, which was then organized and visualized using a linear model (LM) and hierarchical clustering (HCL) with the average linkage clustering method to view the whole dataset by the TIGR multiexperiment viewer (MeV 4.8.1, http://www.tigr.org/software). The BGC distances between different *Bacillus* genomes were then defined as the similarity at the level of GCFs and were directly extracted from the above clustering data. Connection of *Bacillus* genomes with different correlation between BGC distance and phylogenetic distance, was visualized using CIRCOS software (version 0.67-7, http://www.circos.ca).

**Antagonism assay**. The inhibition of antagonistic strains on target strains (listed in Supplementary Data 7) was evaluated by both colony antagonism and fermentation supernatant inhibition assessments. Five milliliters of a diluted overnight culture of each target strain (~$10^5$ CFU mL$^{-1}$) was spread onto LLB plates (10 × 10 cm) to be grown as a bacterial lawn. For the colony antagonism assay, 4 μL of the antagonistic strain culture (~$10^8$ CFU mL$^{-1}$) was spotted onto plates covered by the target strain layer; for the fermentation supernatant inhibition assay, 150 μL of the filtration-sterilized extracellular supernatant of the antagonistic strain cultured in Landy medium for 48 h was injected into an Oxford cup and then placed on lawns of the target strain. The plates were placed at 22 °C until a clear zone formed around the spot, and the inhibition was scored. A heatmap showing the profile of the fermentation supernatant inhibition assay was built by the GraphPad Prism 8.3.0 (https://www.graphpad.com/). Each antagonism assay using wild-type strains includes three biological replicates, while each assay using mutants of strain SQR9 includes six biological replicates.

**B. velezensis SQR9 mutant construction**. Marker-free deletion strains of the target BGC genes were constructed using the strategy previously described by Zhou et al.[51]. In detail, the left franking (LF) region (~1000 bp), direct repeat (DR) sequence (~500 bp), and right flanking (RF) region (~1000 bp) were amplified from genomic DNA of strain SQR9 using the primer pairs LF-F/LF-R, DR-F/DR-R, and RF-F/RF-R, respectively. The PS cassette (~2300 bp; Spc resistance gene) was amplified with the primers PS-F/PS-R using p7S6 as a template[52]. The four fragments were fused using overlap PCR in the order of LF, DR, PS cassette, and RF. Subsequently, the fused fragments were transformed into competent cells of *B. velezensis* SQR9, and the transformants were selected via first-step screening on LLB plates containing Spc. The final mutants were obtained by combining LLB medium containing 10 mM p-Cl-Phe and the verified primer pair VF/VR. The primers used in this study are listed in Supplementary Data 9. Subsequently, each antagonism assay using different mutants includes six biological replicates.

**Statistics**. LM was performed in the R package (version 3.6.1) to assess the correlation between BGC distance and phylogenetic distance, inhibition phenotype and phylogenetic distance/predicted BGC distance, as well as the association of antagonism-phylogeny/BGC-phylogenetic distance/antagonism-BGC distance and the quantity of BGCs in antagonistic strains. Statistical significance was based on a *P*-value < 0.01. Graphs of the correlation data were created using the "ggplot2" package in R; the gray shaded areas denote the 95% confidence intervals. The Duncan's multiple rang tests (*P* < 0.05) of the SPSS version 22.0 (IBM, Chicago, IL, version 22.0) was used for statistical analysis of differences among treatments.

**Reporting summary**. Further information on research design is available in the Nature Research Reporting Summary linked to this article.

## Data availability

The authors declare that the data supporting the findings of this study are available within the article and its Supplementary Information, Supplementary Data or Source Data files, or from the corresponding authors upon request. The DNA sequences from all incubation samples are deposited in the GenBank database with accession number listed in Supplementary Data 7. Source data are provided with this paper.

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

## Acknowledgements
This work was financially supported by the National Natural Science Foundation of China (42090060, 32072665, and 31972512), the Fundamental Research Funds for the Central Universities (KJQN201919), and the Agricultural Science and Technology Innovation Program of CAAS (CAAS-ZDRW202009).

## Author contributions
N.Z., R.Z., and Q.S. conceived and designed the study. L.X. and A.C. performed experimental work. Y.M., L.X., Y.L., Z.L., Z.X., and W.X. analyzed the data. Y.L., X.S., and Y.X. contributed materials. N.Z., R.Z., Y.M., and L.X. wrote the manuscript.

## Competing interests
The authors declare no competing interests.
