## [Peer Review File · Nature Communications]

Reviewers' Comments:

Reviewer #1:

None

Reviewer #2:

Remarks to the Author:

This article by Xia et al. examined the antagonism between multiple strains of the Bacillus genera, first by analyzing the genome of over 4 thousand strains to examine the presence of BGC clusters, and then by looking at direct antagonism between various strains. This study is interesting and includes the analysis of a large amount of data, but it also contains important pitfalls.

1. Many studies (I Mandic-Mulec, Lyons & Kolter, A Kovac, R. Borriss...) were published in the last 10-15 years on intra- and inter-species kin recognition/antagonism in Bacillus and on distribution of BGC. In certain of these studies, the importance (or lack of importance) of BGC in antagonism is evaluated. The article by Xia et al. does not mention any of these studies, and an unfamiliar reader might think that this subject is entirely novel; however certain important aspects of this study were already discussed, though in smaller scale, in previous literature. This literature needs to be integrated and discussed fully in this manuscript.

2. In most of the article, the authors correctly states that absence of biosynthetic gene clusters and phylogenic distance correlates with antagonism - in Figure 2, 3 and 4 correlation is shown, and not causation. Figure 5 looks directly at the importance of BGC and shows that certain BGC contribute up to 40% antagonism according to the graph. While this is an important proportion, it is not sufficient to attribute antagonism to the BGCs and to demonstrate that interference competition was dependent on the BGCs, as concluded in the article.

3. Most strains used in Fig 4 and 5 appears not to have been sequenced (genome not listed in Table S1), and I suppose that presence or absence of BGC is only predicted. While the threshold for prediction is high, because fig 5 is the only one looking at direct involvement of BGC in antagonism, not knowing with certainty the BGC present in the 24 strains makes the conclusion weak. If the genome were examined, it needs to be stated.

Minor comments:

5. Presence of a BGC cluster does not automatically link to production of the molecule. Many clusters have point mutations or regulation motif impairing production of the molecule - this aspect is absent from the manuscript.

6. There are many typos, errors of language and precision missing, but because there is no line number, I cannot provide suggestion of changes.

7. Figure S1 is unreadable.

8. Is there statistical analysis for figure 3?

Reviewer #3:

Remarks to the Author:

The work addresses very interesting and relevant questions, it is hypothesis driven and brings interesting and novel information on BGCs distribution in Bacillus genus. The work is based on bioinformatics analysis and experimental verifications. The main aim was to test the hypothesis stating that more closely related strains/species will share more phylogenetically similar BGCs clusters and thus be less antagonistic toward strains from closely related species (same clade) than towards more distant species of the genus, which carry more different BGCs or lack them. Although the work is interesting and novel I also have some critical comments and suggestions to authors.

Major comments:

- By using antiSMASH and hierarchical clustering on more than 4000 genomes in the data base authors test the correlation between phylogenetic distance of genomes and the pBGCs phylogeny. They obtain a correlation factor $R^2 = 0.2847$, which is not very high correlation for biological systems. Authors should address this result from a critical point of view and discuss it.

- Positive correlation between phylogeny and presence /absence of antibiotic clusters within *B. subtilis* species and closely related species has been shown by Lyon's et al (2016) who analyzed approx. 50 genomes for BGC clusters and found correlation between strength of antagonisms and phylogeny, which is in line with authors hypothesis- however their work addressed within and between closely related species. Later on Lyons and Kolter confirmed this finding on a larger set of strains showing that correlation was lost when more distantly related Bacilli were tested for antagonisms. Xiu et al should comment on these findings.

-

- Recently more in-depth bioinformatics analyses has been performed by Steinke et al, 2021, who analysed more than 310 genomes of *Bacillus subtilis* clade also by antiSMASH- 5 (as in this work) and found that there content of BGCs differs between species of *subtilis* clade. Xiu et al extended bioinformatics analyses further to other clades within genus but I miss a detail analyses within clades so that this work will extend previously published work, which would be useful to better understand the interactions within and between species.

-

- Unfortunately a strong bias of # of genomes in the data base towards *subtilis* and *cereus*, while *megaterium* and *circulans* have significantly fewer # of genomes may skew the results. Authors should be careful with their conclusions and discuss findings presented in Table 3 also from this angle. It is however interesting that they find that *subtilis* genomes harbor higher number of PKSother clusters and PKS-NRPS hybrids than other clades , in contrast RiPPs predominate in *cereus* clade. I miss in the intro section a sentence on potential types of clusters that they expect to find in *Bacillus* genomes to make the reading of the manuscript easier.

- Xia et al performed more in depth analyses on 545 *Bacillus*, however, I am not clear how exactly were 545 genomes chosen nor what is the taxonomic distribution of genomes across clades within this group- Authors should prepare a summary table(supplement) indicating how many strains are available for each species and within each clade- I have not find such a table among supplementary info?

- Xia et al applied BiG-Scape to cluster / identify groups of similar BGC and clustered them into 76 clans, which is visually nicely presented in Fig 2b. However, I miss description which BGC are within one clan. Is it possible to extract this info and prepare a figure /visual representation for a supplementary information? The table S4 is very difficult to understand. Do authors know which and how many specific BGCs are associated with each clan. Also, I wondered why *circulans* has no branches (Fig 2b)? What is represented in the middle of Fig 2 with this very red line- it seems that one clan is present in almost all species and that most of the phylogenetic branches include almost 5 of these BGCs. I miss that this has not been discussed in the text and remains uninformative

-

- Steinke et al find 74 known BGSs clusters in *Bacillus subtilis* clade, authors should use this approach to analyze their data set and compare their findings to previous work.

-

- Authors conclude that due to BGC-phylogeny coherence community assembly will be now much easier as one can predict who will be antagonistic to whom based on BGC homology However, genomics does not reveal whether BGC loci are expressed in specific conditions- it is well known that many loci are silent. Therefore, interactions between strains will still need to be tested experimentally to verify the prediction and consequently successful community assembly, which should be discussed. Besides, correlation between phylogeny and antagonisms is very low in Fig 3 ($R^2 = 0.1263$ and $R^2=0.1132$) and Fig 4 ($R^2 = 0.1598$ and $R^2 = 0.1451$). Authors should discuss this problem and what do the low numbers mean. in Fig 3b- inhibition zone between *pumilus* and *cereus*- which are rather distant (-0.061) do not increase with phylogenetic distance? Is this correlation sufficient to chose strains for community assembly just based on phylogeny?

-

- Xia et al test antagonistic effects in colony confrontation assay against 61 strains representing 4 clades defined by 16S rRNA sequencing. How do they verify which species they are testing if it is

common to find 100% identity between 16S rRNA genes from different species in *Bacillus* genus?

- Xia et al also address specific mechanisms involved in the antagonisms by testing supernatants of 10 BGC mutants of *B. velezensis* SQR9 against 24 strains from 4 clades. They conclude that there is strong correlation between phylogeny and sensitivity to specific antibiotics and that antagonisms is present only if specific antibiotic cluster is missing in the target strain. Although this seems very logical and expected I am not sure the data to support this conclusion are in line with the conclusion. For example, difficidin acts against species in *subtilis*, *circulans* and *megaterium* groups- so it is not only present in one clade? , Also bacillaene contributes to antagonism against many species in *subtilis* and *cereus* clades. This should be addressed and explained. Are genomes of the target strains available to verify this conclusion? – if I understood correctly, authors claim BGC with indicated yellow crosses are present in 80% of the strains indicated by yellow cross and thus are probably not involved in interactions– but in Figure 5 also other genes which seems not be present in at least 80 % of genomes also do not contribute to antagonism? could authors explain this discrepancy

Minor points

Line 47. Sentence is not clear

Line 85: diversiform niches change to diverse niches-

Figure legends 3 and 4: how has statistics been performed; how many biological replicates have been performed for each antagonistic test?

Line 176 add with; coincides WITH the weak inhibition

Biological experiments – supplementary material should contain at least some visual representations of the assays

Table S6 should be organized so that it is clear which strains have been used for which experiment

Type of statistics applied should be added to each figure legend, where relevant

English should be improved- there are still sentences difficult to follow

Response to the Reviewers' comments

Response to Reviewer #1

This article by Xia et al. examined the antagonism between multiple strains of the *Bacillus* genera, first by analyzing the genome of over 4 thousand strains to examine the presence of BGC clusters, and then by looking at direct antagonism between various strains. This study is interesting and includes the analysis of a large amount of data, but it also contains important pitfalls.

Response: Thank you very much for your positive comments. The detailed response to your recommendations and suggestions are as follows.

1. Many studies (I Mandic-Mulec, Lyons & Kolter, A Kovac, R. Borriss...) were published in the last 10-15 years on intra- and inter-species kin recognition/antagonism in *Bacillus* and on distribution of BGC. In certain of these studies, the importance (or lack of importance) of BGC in antagonism is evaluated. The article by Xia et al. does not mention any of these studies, and an unfamiliar reader might think that this subject is entirely novel; however certain important aspects of this study were already discussed, though in smaller scale, in previous literature. This literature needs to be integrated and discussed fully in this manuscript.

Response: Thank you very much for your constructive suggestion. Missing the citation and discussion of several important relevant studies of Mandic-Mulec et al., Lyons & Kolter, Kovac et al., Borriss et al., and so on, is a major drawback of our manuscript. We sincerely realized this important problem and have carefully addressed it in the revised manuscript.

1) We have cited the publications of Lyons & Kolter (Ref. #34), Steinke et al. & Kiesewalter et

al. (Kovacs's laboratory, Ref. #13 & #14), and Chen et al. (Borriss's laboratory, Ref. #16) in the original manuscript; but without sufficient discussion and comparison. In detail, we cited the study of Lyons & Kolter (Ref. #34) in the Discussion part to present the role of public goods utilization in mediating intra- and inter-species interactions, but didn't discuss the relationship between phylogenetic relatedness and kin discrimination/antagonism; we cited the studies by Steinke et al. & Kiesevalter et al. (Ref. #13 & #14) to explain the general importance of biosynthetic gene clusters (BGCs) in bacterial inhibition and immunity, but didn't discuss their detailed phylogenetic distribution in genomes of *Bacillus subtilis* group; we only cited the study by Chen et al. (Ref. #16) to describe the general prevalence of BGCs in *B. amyloliquefaciens*, but didn't discuss their other outstanding contribution on the chemical structure, biological function, and molecular regulation of numbers secondary metabolisms in detail.

2) The kin discrimination in *Bacillus* (Stefanic et al. & Lyons et al. in Mandic-Mulec & Kolter's labs) was not recognized in our manuscript, which is a major defect of this manuscript. We mistakenly neglected the relevance of kin discrimination due to our improper understanding that kin discrimination is something different from antagonism. In Stefanic et al. & Lyons et al. studies, they demonstrated that the interspecies kin discrimination within *B. subtilis* was positively correlated with phylogenetic distance, which was mediated by a collection of genes involved in antimicrobials and cell-surface modifiers. We thought kin discrimination is mainly occurred within (related) species and modulated by close contact (e.g., toxin-antitoxin system and cell surface contact), while interference competition can occur among all bacteria and from longer distance (usually modulated by diffusible secondary antibiotics). Actually, kin discrimination is an important form of

interference competition and have been fully recognized and well discussed in our manuscript (Stefanic et al. & Lyons et al., Ref. #12 & #13 in the revised manuscript).

3) We understand that missing the integration and discussion of these relevant publications is a major defect of our manuscript; we have integrated and fully discussed these literatures (also other relevant publications) in the revised manuscript. In detail, our study concerned the key scientific problem of the correlation between bacterial interference competition and phylogenetic distance, as well as the intrinsic biological mechanism. In the Introduction part, we summarized that the negative correlation among different phyla was mediated by metabolic similarity (Ref. #9 in the revised version), and positive correlation within *Vibrio* or *Streptomyces* was mediated by effect of the prior coexistence or distribution of secondary metabolites, respectively (Ref. #7, 10, 11 in the revised version); in addition, the positive relationship between kin discrimination (or antagonism) and phylogeny within *B. subtilis* was modulated by genes involved in antimicrobials and cell-surface modifiers (Stefanic et al.; Lyons et al.; Kiese-walter et al., Steinke et al.; Ref. #12, #13, #16 and #17 in the revised manuscript, respectively), but this correlation was lost or even became a certain extent of negative with regard to distant *Bacillus* strains (Lyons & Kolter, Ref. #14 in the revised version), which was dependent on the demand of protecting public goods. To summarize, the correlation and driving mechanism are still controversial with regard to strains from different taxonomical scale or groups; specifically, although the relevance of BGCs to antagonism or phylogeny have been evaluated, whether, to what extent, and how does the BGC-phylogeny association itself can affect the relationship between interference competition and phylogenetic distance, have not been well addressed, which limits the understanding of antagonism-phylogeny

correlation from the perspective of diffusible antibiotics (Line 48~84). Based on bioinformatics analysis and experimental verifications, we pronounce that the BGCs similarity-phylogeny coherence does contribute to the positive correlation between congeneric antagonism and phylogenetic distance, and this phenotype is particularly more significant for strains harboring abundant BGCs. In the Discussion part, we compared our finding with the above relevant studies, to demonstrate the different pattern and driving factors of antagonism-phylogeny correlation for interactions on different taxonomical scale (Line 226~274); specifically, we discussed the probable reasons for explaining the discrepancy between Lyons & Kolter's study and our research (Line 275~290). Taken together, we believe these relevant publications do not compromise the novelty of our work, and the most notable merits of our manuscript (BGCs-phylogeny coherence contributes to the positive correlation between congeneric antagonism and phylogenetic distance, and this phenotype is more significant for strains harboring abundant BGCs) is an important advance of these studies.

2. In most of the article, the authors correctly states that absence of biosynthetic gene clusters and phylogenic distance correlates with antagonism - in Figure 2, 3 and 4 correlation is shown, and not causation. Figure 5 looks directly at the importance of BGC and shows that certain BGC contribute up to 40% antagonism according to the graph. While this is an important proportion, it is not sufficient to attribute antagonism to the BGCs and to demonstrate that interference competition was dependent on the BGCs, as concluded in the article.

Response: Thank you very much for your suggestions. As the reviewer commented, the relationship between antagonism and BGCs presence/absence or phylogeny was investigated by correlation analysis in Figs. 2, 3, and 4, and was experimentally verified in Fig. 5. The insufficient contribution of BGC to antagonism (up to 40%) in the original Fig. 5 was attributed to the improper scaleplate in the heatmap. Actually some of the BGCs (e.g., difficidin, macrolactin) contributed up to more than 60% of the inhibition, and disruption of the gene *sfp* (encoding 4'-phosphopantetheinyl transferase that necessary for synthesis of numerous antibiotics) almost completely abolished the antagonistic activity of strain SQR9. We have modified the scaleplate in this figure to range from 0 to 70% for clearly showing the contribution of each BGC.

3. Most strains used in Fig 4 and 5 appears not to have been sequenced (genome not listed in Table S1), and I suppose that presence or absence of BGC is only predicted. While the threshold for prediction is high, because fig 5 is the only one looking at direct involvement of BGC in antagonism, not knowing with certainty the BGC present in the 24 strains makes the conclusion weak. If the genome were examined, it needs to be stated.

Response: Thank you very much for your comments. Only a part of the strains used in Fig. 4 (12 of 42) and Fig. 5 (11 of 25) have been sequenced, and we have noted this information in the revised Table S8. In Fig. 5, for the sequenced strains, we predicted the BGCs presence/absence based on their complete genome; while for the non-sequenced strains, their BGCs presence were predicted based on the corresponding genomes in the same species, with a threshold that more than 80% of the genomes possess the cluster. Considering the conservation of BGCs distribution within the same

species (Fig. 2b & 2c in the manuscript) (Steinke et al., 2021), we think this criterion can reliably predict the BGCs presence/absence in the testing strains.

Minor comments:

4. Presence of a BGC cluster does not automatically link to production of the molecule. Many clusters have point mutations or regulation motif impairing production of the molecule – this aspect is absent from the manuscript.

Response: Thank you for your suggestion. We have discussed this point in the revised Discussion as:

"Furthermore, other potential factors may also contribute the relatively low correlation between antagonism and phylogeny, for example, the unknown secondary metabolites of bacteriocins may specifically kill close relatives, internal genetic variation (e.g., point mutations, partial deletion, altered gene regulation, and silent expression) or external cues (e.g., environmental factors and competing strains) can affect the antibiotic production^{11,17,41,42}, and the undiscovered genetic and physiological features may also regulate the response to different predicted BGCs." (Line 266~271)

5. There are many typos, errors of language and precision missing, but because there is no line number, I cannot provide suggestion of changes.

Response: Sorry for these mistakes and lack of line number. We have carefully gone through the whole manuscript to revise the typos, language errors, precision missing, and other mistakes; we also provided the line number in the revised manuscript.

6. Figure S1 is unreadable.

Response: Sorry for unclarity. We have resubmitted Fig. S1 to ensure it is readable.

7. Is there statistical analysis for figure 3?

Response: Sorry for unclarity. We have supplemented the statistical information (Duncan's multiple range tests) in the revised Fig. 3.

Response to Reviewer #2

The work addresses very interesting and relevant questions, it is hypothesis driven and brings interesting and novel information on BGCs distribution in *Bacillus* genus. The work is based on bioinformatics analysis and experimental verifications. The main aim was to test the hypothesis stating that more closely related strains/species will share more phylogenetically similar BGCs clusters and thus be less antagonistic toward strains from closely related species (same clade) than towards more distant species of the genus, which carry more different BGCs or lack them. Although the work is interesting and novel I also have some critical comments and suggestions to authors.

Response: Thank you very much for your positive comments. The detailed response to your recommendations and suggestions are as follows.

Major comments:

1. By using antiSMASH and hierarchical clustering on more than 4000 genomes in the data base authors test the correlation between phylogenetic distance of genomes and the pBGCs phylogeny. They obtain a correlation factor $R^2 = 0.2847$, which is not very high correlation for biological systems. Authors should address this result from a critical point of view and discuss it.

Response: Thank you very much for your comments. We think this result can be discussed from three aspects: Firstly, the positive correlation indicates that the BGCs distribution pattern is generally in accordance with the phylogenetic relationship. Secondly, in detail, Figs. 2c and S5 showed that the BGCs profile were quite similar within species or between closely related species, as the dots fit well with the regression line when phylogenetic distance < 0.1 ; while the BGCs

distribution between species with distance > 0.25 became highly variable as some dots still matched the line but most of them dispersed around the line. Thirdly, in general, the low R^2 (or not very high) coincides with that BGCs can have a high transferability among different genomes and are usually acquired through horizontal gene transfer (HGT), therefore, the correlation is relatively lower as compared with other conserved genes (e.g., housekeeping genes or metabolism relevant characteristics) (Levy & Borenstein, 2013). The critical point of this result and the related discussion have been included in the revised manuscript (Line 134~140 & Line 299~305), and we also provided an additional figure to show the detailed correlations between genomes with different phylogenetic or BGC distance (Fig. S5 in the revised manuscript).

2. Positive correlation between phylogeny and presence/absence of antibiotic clusters within *B. subtilis* species and closely related species has been shown by Lyon's et al (2016) who analyzed approx. 50 genomes for BGC clusters and found correlation between strength of antagonisms and phylogeny, which is in line with authors hypothesis- however their work addressed within and between closely related species. Later on Lyons and Kolter confirmed this finding on a larger set of strains showing that correlation was lost when more distantly related *Bacilli* were tested for antagonisms. Xia et al should comment on these findings.

Response: Thank you very much for your constructive suggestions. We have cited and carefully discussed these publications in the revised manuscript, introduced the background of relationship between antagonism and phylogeny or BGCs distribution, compared their findings with our data of

this study, and discussed the similarities/differences. These relevant information in the revised manuscript include but not limited to:

"Additionally, the positive relationship between kin discrimination and phylogeny was indicated within *Bacillus subtilis*, which was modulated by genes involved in antimicrobials and cell-surface modifiers^{12,13}; however, this correlation was lost or even became a certain extent of negative when more distantly *Bacillus* strains were tested for antagonism, probably being dependent on the demand of protecting public goods¹⁴." (Line 57~61)

"With regard to the microbial fierce competition exemplified by antagonism, ..., some other scientists discovered a positive relationship between antagonistic interaction (including kin discrimination) and phylogenetic dissimilarity in genus *Vibrio*⁷ and *Streptomyces*^{10,11}, and species *B. subtilis*^{12,13}." (Line 229~232)

"Noticeably, there are some differences between our finding and that reported by Lyons & Kolter, who demonstrated a negative correlation between kin discrimination and phylogeny¹⁴. ... In general, we consider that Lyons & Kolter's study has provided important knowledge with regard to kin discrimination and close contact, especially in a mixed bacterial population¹⁴; and our study focused more on the antagonistic interaction *sensu lato*, which can occur within a longer distance." (Line 275~290)

3. Recently more in-depth bioinformatics analyses has been performed by Steinke et al, 2021, who analysed more than 310 genomes of *Bacillus subtilis* clade also by antiSMASH- 5 (as in this work) and found that the content of BGCs differs between species of *subtilis* clade. Xia et al extended

bioinformatics analyses further to other clades within genus but I miss a detail analyses within clades so that this work will extend previously published work, which would be useful to better understand the interactions within and between species.

Response: Thank you very much for your comments and sorry for missing the detailed analysis.

Here we attached the detailed profile of BGC products and classification to the phylogenetic tree of *Bacillus* genomes (Fig. S3 in the revised manuscript). Together with the biosynthesis gene cluster families (GCFs) distribution among different species (Figs. 2b & S4 in the revised manuscript), we demonstrated that the BGCs profile was consistent with the phylogenetic clade, also different species within clade or even different strains within species, can accommodate their distinct BGCs. For example, *subtilis* clade possessed bacilysin as a specific BGC as compared with other three clades; while within *subtilis* clade, *B. amyloliquefaciens* and *B. velezensis* were the dominant producers of difficidin and macrolactin, and only a certain of strains in this two species can potentially synthesis mersacidin, plantazolicin, and plipastatin. These relevant analysis and discussion has been included in the revised manuscript (Line 107~108, Line 117~131, & Line 291~299).

4. Unfortunately a strong bias of # of genomes in the data base towards *subtilis* and *cereus*, while *megaterium* and *circulans* have significantly fewer # of genomes may skew the results. Authors should be careful with their conclusions and discuss findings presented in Table 3 also from this angle. It is however interesting that they find that *subtilis* genomes harbor higher number of PKSother clusters and PKS-NRPS hybrids than other clades, in contrast RiPPs predominate in

cereus clade. I miss in the intro section a sentence on potential types of clusters that they expect to find in *Bacillus* genomes to make the reading of the manuscript easier.

Response: Thank you very much for your comments and sorry for unclearness. We have downloaded all the available *Bacillus* genomes in NCBI database, resulting in the 4,268 genomes used in the manuscript. It should be note that the number of genomes in *subtilis* or *cereus* clades was indeed much more than those in *megaterium* or *circulans* clades, probably strains in these species have been isolated more frequently and received more attention. Anyway, we think the BGC statistics based on more than 170 genomes in both *megaterium* and *circulans* clades can represent the distribution of secondary metabolites in these species.

In addition, we have presented the information about potential BGC types of *Bacillus* genomes in the revised Introduction part as:

"To test this hypothesis, we referred to the Gram-positive *Bacillus* as the target genus, ... , including non-ribosomal lipopeptides (e.g., surfactin, iturin, and fengycin families produced by various species)²³, non-ribosomal polyketides (e.g., difficidin and macrolactin produced by *B. velezensis*)¹⁹, peptide-polyketide hybrid compound (e.g., zwittermicin produced by *B. cereus* and *B. thuringiensis*)²⁴, and ribosomally synthesized and post-translationally modified peptides (RiPPs; e.g., lichenicidin produced by *B. licheniformis*)²⁶." (Line 73~79)

5. Xia et al performed more in depth analyses on 545 *Bacillus*, however, I am not clear how exactly were 545 genomes chosen nor what is the taxonomic distribution of genomes across clades within this group- Authors should prepare a summary table(supplement) indicating how many strains are

available for each species and within each clade- I have not find such a table among supplementary info?

Response: Thank you very much for your comments and sorry for unclearness. The 545 *Bacillus* genomes were chosen based on: (i) high genome sequencing quality for further BGC distance calculation, and (ii) covering all *Bacillus* species. All these criteria have been included in the revised manuscript (Line 110~112). As a result, totally 545 representative genomes covering 137 species were chosen for further analysis. In addition, we have provided a Supplementary Table S2 to indicate the quantity of available genomes for each species and within each clade.

6. Xia et al applied BiG-Scape to cluster / identify groups of similar BGC and clustered them into 76 clans, which is visually nicely presented in Fig 2b. However, I miss description which BGC are within one clan. Is it possible to extract this info and prepare a figure /visual representation for a supplementary information? The table S4 is very difficult to understand. Do authors know which and how many specific BGCs are associated with each clan. Also, I wondered why circulans has no branches (Fig 2b)? What is represented in the middle of Fig 2 with this very red line- it seems that one clan is present in almost all species and that most of the phylogenetic branches include almost 5 of these BGCs. I miss that this has not been discussed in the text and remains uninformative.

Response: Thank you very much for your constructive suggestions and sorry for unclearness. In the original manuscript, we calculated the (dis)similarity of paired BGCs and assigned them into 1,110 gene cluster families (GCFs); next, based on calculation of the distance between different GCFs, we further distributed them into 76 gene cluster clans (GCCs). It can be understood that GCCs are a

higher classification of GCFs. However, as clans were obtained based on GCF distance but not the detailed structural characteristics of BGCs in each family, we noted that some families including BGCs with very different chemical structure, can be clustered into the same clan; for example, the very red line in original Fig. 2b contained numerous GCFs but not all of them were actually similar or relevant. Therefore, at least in our study, GCFs are more appropriate to describe the BGCs distribution pattern than GCCs, and we then re-prepared the hierarchal clustering of the 545 *Bacillus* genomes based on statistics of their GCFs. It can be seen that in the revised Fig. 2b, the distribution of BGCs is more clear and matches well with the phylogenetic clades of *Bacillus*, where genomes in *circulans* clade also contained different branches. We have included a color stripe in the top of the heatmap to show the BiG-SCAPE class of each GCF. We also provided a larger version of this figure as a supplementary material (Fig. S4), which shows the detailed information of all testing genomes and the BGC products in each family. In addition, besides the information of GCFs and GCCs, we have annotated the predicted product and BiG-SCAPE class of each BGC listed in the revised Table S5 (original Tables S4); we also supplemented an additional Table S6 to explain which GCF (as well as their product and BiG-SCAPE class) are included in each BGC clan, as well as their corresponding BGC product and classification.

7. Steinke et al find 74 known BGCs clusters in *Bacillus subtilis* clade, authors should use this approach to analyze their data set and compare their findings to previous work.

Response: Thank you very much for your suggestions. Actually, we applied the same approach in the manuscript to predict the BGCs in the testing *Bacillus* genomes, and identified 117 BGC

products, 310 GCFs, and 47 GCCs in the *B. subtilis* clade (include but not limited to the species analyzed in the study of Steinke et al.; Table S5 in the revised manuscript). Generally, the types and distribution of BGCs are in accordance with the finding in the previous work. We have described these data and discussed this point in the revised manuscript (Line 123~127 & Line 295~299).

8. Authors conclude that due to BGC-phylogeny coherence community assembly will be now much easier as one can predict who will be antagonistic to whom based on BGC homology. However, genomics does not reveal whether BGC loci are expressed in specific conditions- it is well known that many loci are silent. Therefore, interactions between strains will still need to be tested experimentally to verify the prediction and consequently successful community assembly, which should be discussed. Besides, correlation between phylogeny and antagonisms is very low in Fig 3 ($R^2 = 0.1263$ and $R^2=0.1132$) and Fig 4 ($R^2 = 0.1598$ and $R^2 = 0.1451$). Authors should discuss this problem and what do the low numbers mean. in Fig 3b- inhibition zone between *pumilus* and *cereus*- which are rather distant (-0.061) do not increase with phylogenetic distance? Is this correlation sufficient to choose strains for community assembly just based on phylogeny?

Response: Thank you very much for your constructive comments and suggestions. Silent expression, as well as point mutations, partial deletion, and variable regulation motif, are all common reasons causing impaired production of secondary metabolites encoded by BGCs in the genome, which can influence the correlation between observed antagonism and phylogeny; we have discussed this point in the revised manuscript as "Furthermore, other potential factors may also contribute the relatively low correlation between antagonism and phylogeny, for example, the unknown secondary

metabolites of bacteriocins may specifically kill close relatives, internal genetic variation (e.g., point mutations, partial deletion, altered gene regulation, and silent expression) or external cues (e.g., environmental factors and competing strains) can affect the antibiotic production^{11,17,41,42}, and the undiscovered genetic and physiological features may also regulate the response to different predicted BGCs.” (Line 266~271).

On the other hand, we found that for each individual antagonistic strain, the correlation between antagonism and phylogeny, was also positively associated with the BGCs No. in the genome (Fig. 4d in the revised manuscript). It means that antagonistic strains carrying abundant BGCs tend to announce a strong positive correlation between inhibition phenotype and phylogenetic distance, while those with fewer BGCs usually show a weak or even irregular antagonistic pattern against diverse target strains (their unique BGCs confronting targets were quite rare and contributed to a irregular inhibition pattern against different strains) (Fig. 4e & Table S9 in the revised manuscript). This pattern can partially explain the relative low antagonism-phylogeny correlation observed in the fermentation supernatant assay, which has been discussed in the revised manuscript (Line 176~188 & Line 255~266).

In Fig. 3B, the inhibition zone of *B. pumilus* ACCC04450 on distant *cereus* clade (average phylogenetic distance of 0.061) was indeed slighter than that on closer *megaterium* clade (~0.055); while the antagonism toward *pumilus* subclade (~0.002), *subtilis* subclade (~0.030), *circulans* clade (~0.053), and *megaterium* clade (~0.055) gradually increased with phylogenetic distance. We demonstrated that in general, the congeneric antagonism reveals to be stronger toward distant species, although there could be some exceptions caused by different biological mechanisms (as

discussed above). We think this correlation can provide implications for predicting interactions among strains with different phylogenetic relationship, especially for those possessing abundant BGCs; anyway, the actual community assemblage process still needs experimental verifications.

9. Xia et al test antagonistic effects in colony confrontation assay against 61 strains representing 4 clades defined by 16S rRNA sequencing. How do they verify which species they are testing if it is common to find 100% identity between 16S rRNA genes from different species in *Bacillus* genus?

Response: Thank you very much for your comments. Species of the testing strains were assigned based on the closest match to 16S rRNA gene sequences through the EzBioCloud and NCBI databases; for strains showing 100% identity of 16S rRNA gene with different *Bacillus* species, we will further check its classification by morphological, physiological, and biochemical identification.

10. Xia et al also address specific mechanisms involved in the antagonisms by testing supernatants of 10 BGC mutants of *B. velezensis* SQR9 against 24 strains from 4 clades. They conclude that there is strong correlation between phylogeny and sensitivity to specific antibiotics and that antagonisms is present only if specific antibiotic cluster is missing in the target strain. Although this seems very logical and expected I am not sure the data to support this conclusion are in line with the conclusion. For example, difficidin acts against species in *subtilis*, *circulans* and *megaterium* groups- so it is not only present in one clade? Also bacillaene contributes to antagonism against many species in *subtilis* and *cereus* clades. This should be addressed and explained. Are genomes of the target strains available to verify this conclusion? – if I understood correctly, authors claim BGC

with indicated yellow crosses are present in 80% of the strains indicated by yellow cross and thus are probably not involved in interactions– but in Figure 5 also other genes which seems not be present in at least 80 % of genomes also do not contribute to antagonism? could authors explain this discrepancy.

Response: Thank you very much for your comments. The mutants-based experiments demonstrated all antagonism was attributed to the BGCs that specifically present in the antagonistic strain but absent in the target strain, that is the secondary metabolites shared by both strains were not involved in the inhibition; on the other hand, it should be note that not all of the unique BGCs in antagonistic strains will contribute to inhibition of the targets. In detail, difficidin was only existed in *B. velezensis* (*subtilis* clade) but absent in all of the target strains; it can contribute to the antagonism of *subtilis* (partial strains), *circulans*, and *megaterium* clades, but was not involved in the inhibition of *cereus* clade (although they cannot produce difficidin). Bacillaene was present in *B. velezensis* and *B. halotolerans* (*subtilis* clade); it acted against partial species in *cereus* and *circulans* clades but was ineffective towards *subtilis* or *megaterium* clades (although most of them cannot produce this molecule). Whether a target strain being resistant or sensitive to a specific secondary metabolite that it doesn't synthesis, shall involve complicated molecular mechanisms (e.g., the resistance maybe dependent on the production of analogous antibiotics) and still needs further investigation; anyway, the possessed BGCs in the target strain can almost assure the tolerance to this molecule produced by the attacker. In general, this finding verifies that strains sharing fewer BGCs will have a higher probability to inhibit each other. The relevant explanation has been included in the revised manuscript (Line 220~223).

Minor points

1. Line 47. Sentence is not clear

Response: Sorry for unclarity. This sentence has been rewritten as "Phylogenetic relatedness is considered to be closely associated with microbial competition, ...".

2. Line 85: diversiform niches change to diverse niches-

Response: Thanks for your comments. It has been revised as suggested.

3. Figure legends 3 and 4: how has statistics been performed; how many biological replicates have been performed for each antagonistic test?

Response: Sorry for unclarity. For the boxplots in Fig. 3, the Duncan's multiple range tests ($P < 0.05$) were used for analyzing the antagonism data toward strains from different clades, and this statistical information has been included in the revised Fig. 3; for the regression in Fig. 3 & Fig. 4, linear models were performed to assess the correlation between different variates, and both the adjusted R^2 and P -value were shown in the figures.

Each antagonistic test in Figs. 3 & 4 contained three biological replicates, and the mean value was shown in the figures and used for further analysis; this information has been included in the revised manuscript. In addition, Each antagonistic assay in Fig. 5 included six biological replicates, and the average contribution of each BGC was calculated and shown in the heatmap; this

information has been included in the revised manuscript.

4. Line 176 add with; coincides WITH the weak inhibition

Response: Thanks for your comments. It has been revised as suggested.

5. Biological experiments – supplementary material should contain at least some visual representations of the assays.

Response: Thanks for your comments. We have provided a supplementary figure (Fig. S6) to show the visual representations of both colony confrontation and fermentation supernatant assays.

6. Table S6 should be organized so that it is clear which strains have been used for which experiment

Response: Thanks for your comments and sorry for unclearness. We have reorganized Table S8 in the revised manuscript (Table S6 in the original version) to clear show that which strain have been used for which experiment.

7. Type of statistics applied should be added to each figure legend, where relevant English should be improved- there are still sentences difficult to follow

Response: Thanks for your comments and sorry for unclearness. The statistic information has been included in the legend of each figure, also the relevant English has been improved to be more understandable.

References

- Stefanic, P., Kraigher, B., Lyons, N.A., Kolter, R. & Mandic-Mulec, I. Kin discrimination between sympatric *Bacillus subtilis* isolates. PNAS 112, 14042-14047 (2015).
- Steinke, K., Mohite, O.S., Weber, T. & Kovacs, A.T. Phylogenetic distribution of secondary metabolites in the *Bacillus subtilis* species complex. mSystems 6 (2021).
- Levy, R. & Borenstein, E. Metabolic modeling of species interaction in the human microbiome elucidates community-level assembly rules. Proc Natl Acad Sci U S A (2013).
- Lyons, N.A., Kraigher, B., Stefanic, P., Mandic-Mulec, I. & Kolter, R. A combinatorial kin discrimination system in *Bacillus subtilis*. Curr Biol 26, 733-742 (2016).

Reviewers' Comments:

Reviewer #1:

None

Reviewer #2:

Remarks to the Author:

The authors have mostly answered my questions, and did the appropriate changes to the manuscript. However, I really have trouble understanding Fig S5 - more extensive explanations would help people not familiar with this analysis to better understand.

Reviewer #3:

Remarks to the Author:

Xia et al is a comprehensive study involving bioinformatic analyses of publicly available *Bacillus* genomes and laboratory test addressing antagonisms between bacteria. The work shows that a biosynthetic gene cluster (BGCs) phylogeny coherence contributes to the positive correlation between congeneric antagonism and phylogenetic distance, and that this phenotype is more significant for strains harboring abundant BGCs.

Authors have extensively addressed all my comments in their rebuttal and provided all additional analyses requested, revised figures and improved introduction and discussion.

I have only minor comments

Fig 4a _ numbers below the color indicator are shifted?

B. ginsengihumi ACCC05679 is highly antagonistic against all tested strains. If authors have a comment on this it would be useful to include it.

Lines 137_144

I find the paragraph between 137-144 very difficult to follow. I would suggest to paraphrase it. A potential suggestion would be:

"Genomes of phylogenetically close relatives (phylogenetic distance $<0,1$) carried highly similar arsenal of BGCs (Fig 2c, Fig S5). In more distantly related species (phylogenetic distance $>0,3$) the BGCs distance increased with phylogenetic distance but some BGCs were also highly dispersed (Fig 2, Fig S5). This suggests that some BGC loci are shared between different clades (Fig S5), for example, between *subtilis*-*cereus* clades (Fig. S5d) or *circulans* and other clades (Fig. S5h)"

Line 227-229

The sentence is difficult to follow. I would suggest that it is slightly revised ("It should be noted that not all the unique BGCs in antagonistic strains contributed to inhibition of the targets (Fig. 5) and the outcome may also depend on whether a specific secondary metabolite is being synthesized or not and this aspect still needs further investigation.

Reviewer #4:

Remarks to the Author:

The aim of this really nice paper was to characterize and disentangle mechanisms driving the relationship between phylogeny and interference competition among *Bacillus* congeners. Using a combination of bioinformatics and experiments, the authors show that the intensity of inhibition scales with increasing phylogenetic distance, although somewhat weakly, as does the similarity of the BGC in each strain. They also nicely show experimentally with knock-out strains that at least some of the inhibition can be causally associated with specific BGC. The paper has apparently already gone through a comprehensive round of review that led to many changes from the original, and which already addressed some of the aspects that I might have mentioned. Aside

from fairly minor quibbles below, there is little to criticize. This is a big effort that undoubtedly leaves many questions unanswered, as will any study of this scope. However, it is also undoubtedly an interesting big step forward that illuminates the enormous diversity of secondary metabolism in this ubiquitous bacterial group.

- 1) The clustering into 4 main groups is somewhat arbitrary and could be better rationalized, especially the decision to not further partition the analyses of e.g. Fig 2A into subclades or finer levels within the subtilus clade.
- 2) Although the focus is across clades, there seems a missed opportunity to examine the if the same relationships hold within clades where overall phylogenetic relationships are closer. This is done, somewhat, in later analyses where the authors focus on more and less related strains; but this is confounded by the fact that the number of BGCs vary in different clades. In other words, does antagonism scale with phylogeny if you only focus on e.g the subtilis group, etc?
- 3) The number of BGC is significantly lower for the cirulans clade.
 - a. Any idea why this is the case?
 - b. Could this be an artifact of the fact that this group is less represented overall or due to a detection bias of anti-SMASH?
- 4) The justification for choosing the 545 strains needs to be clearer. If these have the best sequences, what does this say about the other ~4000 strains? It should at least be clear that the 545 strains are actually representative (i.e. statistically indistinguishable) from the clades from which they were chosen.
- 5) Same question as above for the 8 focal strains and 61 targets used from Line 152. It's fine to choose a subset—they just need to be rationalized a bit more clearly.
- 6) The data in Figure 5 are a bit difficult to interpret given the color scale. We can see that some BGC matter more than others, but not whether the totality of BGC can explain everything. Would it be possible to add a final column that sums the different contributions of the BGC? This might be complicated given the interactions between BGC that contributed to the zone of inhibition.
- 7) Horizontal gene transfer is often mentioned but not at all considered. Is there any evidence for HGT for any of the BGC, e.g. a lack of concordance between the BGC and the overall phylogeny? If so, these might be worth considering separately as their relationship with the species phylogeny will differ from the other BGC whose evolution is concordant with the species tree.

Response to Reviewer #2

The authors have mostly answered my questions, and did the appropriate changes to the manuscript. However, I really have trouble understanding Fig S5 - more extensive explanations would help people not familiar with this analysis to better understand.

Response: Thank you very much for your positive comments and sorry for unclearness about the information of Fig. S5. We have improved the Figure to make it more intuitive, and also detailed the explanation of this Figure in the revised manuscript to help the readers to understand this analysis more easily as:

"Fig. S5 Connection of *Bacillus* genomes in 7 groups showing different correlation between biosynthetic gene clusters (BGCs) distance and phylogenetic distance. (a) The correlation between the BGC distance and phylogenetic distance of the 545 representative *Bacillus* genomes was showed as a dot plot, in which 7 high-density distinguishable dot areas forms (group 1~7, with 19151 (12.6%), 10787 (7.1%), 31198 (20.4%), 15127 (9.9%), 19313 (12.7%), 16668 (10.9%), and 9026 (5.9%) dots, respectively). (b) For detail analysis, the points of the 7 groups in (a), corresponding to the relationships of each two different *Bacillus* genomes, were extracted, respectively, and shown as the line directly on the phylogenetic tree of the 545 *Bacillus* genomes (b). Both semicircles consists of the 545 *Bacillus* genomes with the same order as in the phylogenetic tree (e.g., group 1 in (b)); each line indicates that the BGC distance and phylogenetic distance of the connected pairwise genomes, represent a dot in the corresponding area/group in (a). Different colors were used to indicate each *Bacillus* clades/subclades: red, *cereus* clade; blue, *pumilus* subclade;

green, *subtilis* subclade; yellow, *megaterium* clade; gray, *circulans* clade. Pd: phylogenetic distance;

BGCd: BGC distance." (Line 664~676)

Response to Reviewer #3

Xia et al is a comprehensive study involving bioinformatic analyses of publicly available *Bacillus* genomes and laboratory test addressing antagonisms between bacteria. The work shows that a biosynthetic gene cluster (BGCs) phylogeny coherence contributes to the positive correlation between congeneric antagonism and phylogenetic distance, and that this phenotype is more significant for strains harboring abundant BGCs.

Authors have extensively addressed all my comments in their rebuttal and provided all additional analyses requested, revised figures and improved introduction and discussion.

I have only minor comments

Response: Thank you very much for your positive comments. The detailed response to your recommendations and suggestions are as follows.

1) Fig 4a _ numbers bellow the color indicator are shifted?

Response: Thank you for your comments and sorry for the mistake. The color bar in Fig. 4a was mistakenly marked in the last version and has been corrected in the revised manuscript.

2) *B. ginsengihumi* ACCC05679 is highly antagonistic against all tested strains. If authors have a comment on this it would be useful to include it.

Response: Thank you for your suggestion. We have commented on this phenomenon in the revised manuscript as:

"Interestingly, *B. ginsengihumi* ACCC05679 was found to be highly antagonistic against

nearly all tested strains (Fig. 4a), which might be attributed to the synthesis of chejuenolide based on prediction using the available conspecific genomes (Table S3), a macrocyclic tetraene that can inhibit diverse Gram-positive bacteria". (Line 182~185)

3) Lines 137_144 I find the paragraph between 137-144 very difficult to follow. I would suggest to paraphrase it. A potential suggestion would be:

"Genomes of phylogenetically close relatives (phylogenetic distance < 0.1) carried highly similar arsenal of BGCs (Fig 2c, Fig S5). In more distantly related species (phylogenetic distance > 0.3) the BGCs distance increased with phylogenetic distance but some BGCs were also highly dispersed (Fig 2, Fig S5). This suggests that some BGC loci are shared between different clades (Fig S5), for example, between *subtilis-cereus* clades (Fig. S5d) or *circulans* and other clades (Fig. S5h)"

Response: Thank you very much for your recommendation and sorry for unclarity. This paragraph has been revised as suggested.

4) Line 227-229 The sentence is difficult to follow. I would suggest that it is slightly revised ("It should be noted that not all the unique BGCs in antagonistic strains contributed to inhibition of the targets (Fig. 5) and the outcome may also depend on whether a specific secondary metabolite is being synthesized or not and this aspect still needs further investigation.

Response: Thank you very much for your recommendation and sorry for unclarity. This sentence has been revised as suggested.

Response to Reviewer #4

The aim of this really nice paper was to characterize and disentangle mechanisms driving the relationship between phylogeny and interference competition among *Bacillus* congeners. Using a combination of bioinformatics and experiments, the authors show that the intensity of inhibition scales with increasing phylogenetic distance, although somewhat weakly, as does the similarity of the BGC in each strain. They also nicely show experimentally with knock-out strains that at least some of the inhibition can be causally associated with specific BGC. The paper has apparently already gone through a comprehensive round of review that led to many changes from the original, and which already addressed some of the aspects that I might have mentioned. Aside from fairly minor quibbles below, there is little to criticize. This is a big effort that undoubtedly leaves many questions unanswered, as will any study of this scope. However, it is also undoubtedly an interesting big step forward that illuminates the enormous diversity of secondary metabolism in this ubiquitous bacterial group.

Response: Thank you very much for your positive comments. The detailed response to your recommendations and suggestions are as follows.

1) The clustering into 4 main groups is somewhat arbitrary and could be better rationalized, especially the decision to not further partition the analyses of e.g. Fig 2A into subclades or finer levels within the *subtilis* clade.

Response: Thank you for your comments. The clustering of the four clades was established on the four dominant branches based on the maximum likelihood phylogenetic tree in Fig. 1 and Fig. S1,

where the latter can show the divergence of the four groups more visually. Since the *cereus*, *megaterium*, and *circulans* clades didn't show clear dominant branches within clade, no subclades were further introduced; while the *subtilis* and *pumilus* subclade were established based on their distinct clustering within *subtilis* clade. Additionally, a Figure showing the statistics of BGCs in the five (sub)clades (*subtilis* subclade, *pumilus* subclade, *cereus* clade, *megaterium* clade, and *circulans* clade) was included as an supplementary graph (Fig. S2a) in the revised manuscript.

2) Although the focus is across clades, there seems a missed opportunity to examine the if the same relationships hold within clades where overall phylogenetic relationships are closer. This is done, somewhat, in later analyses where the authors focus on more and less related strains; but this is confounded by the fact that the number of BGCs vary in different clades. In other words, does antagonism scale with phylogeny if you only focus on e.g the *subtilis* group, etc?

Response: Thank you for your comments. The correlation analysis in the present study (BGC-phylogenetic distance (Fig. 2c), antagonism-phylogenetic distance (Figs. 3e & 4b), antagonism-BGC distance (Figs. 3f & 4c)) was performed for examining the relationship between all tested genomes/strains, covering those both among and within clades and without bias.

Your suggestion for specifically investigating the correspondence within clade is very constructive. We calculated the correlation between antagonism and phylogenetic distance within the four clades, respectively, observing that the significant positive relationship was only obtained in *subtilis* clade, which might be due to the high relative genetic similarity within clade and/or

variation of BGCs abundance in different clades. This result was included in the revised manuscript (Fig. S7).

3) The number of BGC is significantly lower for the *circulans* clade.

a. Any idea why this is the case?

b. Could this be an artifact of the fact that this group is less represented overall or due to a detection bias of anti-SMASH?

Response: Thank you for your comments. Firstly, with regards to the data acquisition, we downloaded all the available *Bacillus* genomes in NCBI database, resulting in the 4,268 genomes used in the manuscript. The number of genomes in *circulans* clade was indeed less than those in *subtilis* or *cereus* clades, probably strains in these species have been isolated less frequently and received less attention. Anyway, we think the analysis based on 181 genomes in *circulans* clade can adequately represent the distribution of secondary metabolites in these species.

Secondly, the BGCs predication was performed based on antiSMASH 5.0 software, the most widely used tool for identifying and analyzing BGCs in bacterial and fungal genome sequences and is regarded as the gold standard (Blin et al., 2019; Blin et al., 2021). antiSMASH integrates and cross-links with a large number of in silico secondary metabolite analysis tools that have been published earlier, and is powered by several open source tools such as NCBI BLAST+, HMMer 3, Muscle 3, FastTree, PySVG and JQuery SVG (<https://antismash.secondarymetabolites.org/#!/about>). Since its initial release in 2011, antiSMASH has been applied to annotate BGCs in a large number of diverse genomes by worldwide researchers as well as BGC-oriented databases (Medema et al.,

2011; Crits-Christoph et al., 2018; Carrión et al., 2019; Sugimoto et al., 2019; Culp et al., 2020; Blin et al., 2021); it has also incorporated in an ecosystem of a number of independent tools such as BiG-SCAPE, ARTS, Pep2Path, and so on (Navarro-Muñoz et al., 2020; Blin et al., 2021).

Based on the two arguments, we consider that both the genomes acquisition and BGCs annotation are representative and comprehensive, thus the *circulans* clade is likely to actually accommodate less BGGs than other *Bacillus* species, suggesting a distinct physiological feature and niche adaptation strategy. Anyway, exploring novel secondary metabolites in *circulans* clade in the future is significant for both expanding our knowledge about BGCs and acquiring active compounds with unique biological functions.

4) The justification for choosing the 545 strains needs to be clearer. If these have the best sequences, what does this say about the other ~4000 strains? It should at least be clear that the 545 strains are actually representative (i.e. statistically indistinguishable) from the clades from which they were chosen.

Response: Thank you for your comments and sorry for unclearness. In detail, the justification for choosing the 545 *Bacillus* genomes includes: (i) high genome sequencing quality that available for the further BGC distance calculation using BiG-SCAPE; (ii) covering all *Bacillus* species in the 4,268 *Bacillus* genomes; (iii) for each species, the representative genome(s) was/were chosen from each of the main branches in the phylogram of genomes within this species. The criteria has been included in the revised manuscript (Line 114~116).

5) Same question as above for the 8 focal strains and 61 targets used from Line 152. It's fine to choose a subset—they just need to be rationalized a bit more clearly.

Response: Thank you for your comments and sorry for unclearness. The 8 focal strains were chosen from *subtilis* and *cereus* clades, which are the two dominant groups within the genus *Bacillus* (David Alcaraz et al., 2010) and accommodate abundant BGCs (Fig. 2a). In detail, *B. amyloliquefaciens* ACCC19745 and *B. subtilis* NCIB 3610 belong to the *subtilis* subclade in *subtilis* clade, and *B. pumilus* ACCC04450 and *B. xiamenensis* TL9 belong to the *pumilus* subclade in *subtilis* clade; while *B. thuringiensis* YX7 is a typical and widespread species in *cereus* clade (Lin et al., 2021; Fig. S1), and *B. mobilis* XL40, *B. bingmayongensis* KF27, and *B. proteolyticus* TZ4 represent species that locating in the main branches within *cereus* clade (Fig. S1). Furthermore, the 61 target strains were picked up based on their representation of the dominant clusters in the *Bacillus* species tree (Fig. S1), resulting in 17 strains in *subtilis* subclade, 15 in *pumilus* subclade, 12 in *cereus* clade, 9 in *megaterium* clade, and 8 in *circulans* clade. Therefore, we think the raised 8 focal strains and 61 targets can be recognized as a representative subset for examining the interaction features within the genus *Bacillus*. These justification has been explained in the revised manuscript (Line 154~159).

6) The data in Figure 5 are a bit difficult to interpret given the color scale. We can see that some BGC matter more than others, but not whether the totality of BGC can explain everything. Would it be possible to add a final column that sums the different contributions of the BGC? This might be

complicated given the interactions between BGC that contributed to the zone of inhibition.

Response: Thank you for your suggestion and sorry for unclearness. We have added a final column in the revised Fig. 5 to show the sum of contribution by different BGCs; it can be seen that the observed inhibition was almost entirely attributed to the listed BGCs.

7) Horizontal gene transfer is often mentioned but not at all considered. Is there any evidence for HGT for any of the BGC, e.g. a lack of concordance between the BGC and the overall phylogeny? If so, these might be worth considering separately as their relationship with the species phylogeny will differ from the other BGC whose evolution is concordant with the species tree.

Response: Thank you for your comments. Correspondingly, we examined the correlation between BGC similarity and phylogenetic distance in all distinct BGC families from the 545 representative *Bacillus* genomes. BGCs in the same BGC family identified by BiG-SCAPE software were considered as homologous, and thus are suitable for horizontal gene transfer (HGT) analysis in different *Bacillus* species. For detail, pairwise distance between BGCs within each family was calculated by combining three similarity scores from Jaccard Index (JI), Adjacency Index (AI), and Domain sequence similarity (DSS) analyses, and then matched to the phylogenetic distance of their derived *Bacillus* genomes. For each analysis (JI, DSS, or AI), the correlations from all BGC families were merged together, and those points with phylogenetic distance ≥ 0.2 (indicating the pairwise genomes are from different clades or at least distant species within clade) and similarity scores ≥ 0.4 (indicating the pairwise genomes accommodate homologous BGCs) were possible consequence from HGT events. As a result, recovering of the points matching the above conditions

(phylogenetic distance ≥ 0.2 & similarity score ≥ 0.4 ; Fig. S9 in the revised manuscript) from either of the three analysis suggested the occurrence of HGT of BGCs among different species in *Bacillus*. This finding has been introduced in the Discussion part of the revised manuscript (Line 256~259 & Line 318~321).

References

- Blin, K. *et al.* antiSMASH 5.0: updates to the secondary metabolite genome mining pipeline. *Nucleic Acids Res* **47**, W81-W87 (2019).
- Blin, K. *et al.* antiSMASH 6.0: improving cluster detection and comparison capabilities. *Nucleic Acids Res* **49**, W29-W35 (2021).
- Carrión, V. J. *et al.* Pathogen-induced activation of disease-suppressive functions in the endophytic root microbiome. *Science* **366** 606-612 (2019).
- Crits-Christoph, A., Diamond, S., Butterfield, C.N., Thomas, B.C. & Banfield, J.F. Novel soil bacteria possess diverse genes for secondary metabolite biosynthesis. *Nature* **558**, 440-444 (2018).
- Culp, E. J. *et al.* Evolution-guided discovery of antibiotics that inhibit peptidoglycan remodelling. *Nature* **578** 582-587 (2020).
- David Alcaraz *et al.* Understanding the evolutionary relationships and major traits of *Bacillus* through comparative genomics. *BMC Genomics* **11** 332 (2010).
- Lin, Y. *et al.* Adaptation of *Bacillus thuringiensis* to plant colonization affects differentiation and toxicity. *mSystems* **6** e00864-21 (2021).

Medema, M. H. *et al.* antiSMASH: rapid identification, annotation and analysis of secondary metabolite biosynthesis gene clusters in bacterial and fungal genome sequences. *Nucleic Acids Res* **39**, W339-W346 (2011).

Navarro-Muñoz, J.C. *et al.* A computational framework to explore large-scale biosynthetic diversity. *Nat Chem Biol* **16**, 60-68 (2020).

Sugimoto, Y. *et al.* A metagenomic strategy for harnessing the chemical repertoire of the human microbiome. *Science* **366** eaax9176 (2019).

Reviewers' Comments:

Reviewer #4:

Remarks to the Author:

The authors have done a very thorough job responding to my few comments and concerns. This is an interesting paper that will motivate quite a lot of follow-up work in this genus.

Response to Reviewer #4

The authors have done a very thorough job responding to my few comments and concerns. This is an interesting paper that will motivate quite a lot of follow-up work in this genus.

Response: Thank you very much for your positive comments. We will continue to motivate in depth in the relevant area.